# Preserving Privacy Through DeMemorization: An Unlearning Technique For Mitigating Memorization Risks In Language Models

**Aly M. Kassem** ◇    **Omar Mahmoud** ♣    **Sherif Saad** ◇

◇School of Computer Science, University of Windsor

♣Applied Artificial Intelligence Institute, Deakin University

◇{kassem6,sherif.saad}@uwindsor.ca

♣o.mahmoud@research.deakin.edu.au

## Abstract

Large Language models (LLMs) are trained on vast amounts of data, including sensitive information that poses a risk to personal privacy if exposed. LLMs have shown the ability to memorize and reproduce portions of their training data when prompted by adversaries. Prior research has focused on addressing this memorization issue and preventing verbatim replication through techniques like knowledge unlearning and data pre-processing. However, these methods have limitations regarding the number of protected samples, limited privacy types, and potentially lower-quality generative models. To tackle this challenge more effectively, we propose "DeMem," a novel unlearning approach that utilizes an efficient reinforcement learning feedback loop via proximal policy optimization. By fine-tuning the language model with a negative similarity score as a reward signal, we incentivize the LLMs to learn a paraphrasing policy to unlearn the pre-training data. Our experiments demonstrate that DeMem surpasses strong baselines and state-of-the-art methods in terms of its ability to generalize and strike a balance between maintaining privacy and LLM performance.

## 1 Introduction

Large language models (LLMs) have experienced exponential growth in recent years, scaling up from millions to billions to trillions of parameters (Radford et al., 2019; Brown et al., 2020; Chowdhery et al., 2022; Fedus et al., 2021). As their scale increases, the training sets for these models also expand to billions of tokens (Gao et al., 2020), leading to overall performance improvements, even in few-shot learning scenarios (Brown et al., 2020). However, this growth in model size and training data has raised practical concerns regarding privacy risks associated with memorizing the training data. Adversaries can extract individual sequences from a pre-trained model, even if the training dataset is publicly available (Carlini et al., 2021).

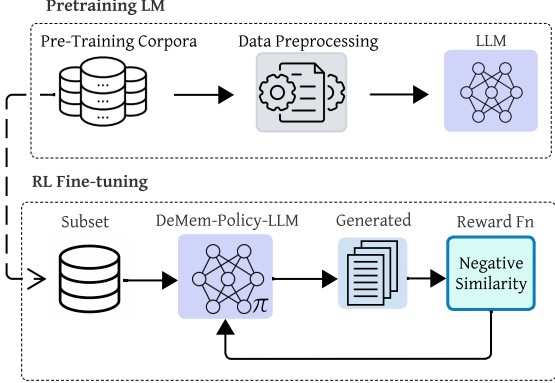

Figure 1: First, LLM is pre-trained on large corpora in which Deduplication is applied. Then, a subset of training corpora is employed to learn the LM a DeMem Policy via negative similarity feedback.

Studies have shown that a language model with 6 billion parameters (GPT-J) can memorize at least 1% of its training data (Carlini et al., 2022). One potential cause of this memorization is the training strategy of the language model, as its objective is to identify the relationships between tokens, either in an auto-regressive LM setup or through masked language modelling (MLM) (Devlin et al., 2018), where the model predicts the masked tokens based on their surrounding context (Radford et al., 2018). Additionally, repeated instances in the training corpus can contribute to memorization, as more frequent examples are more likely to be memorized (Lee et al., 2021). To address the issue of memorization in LLMs, several approaches have been proposed, including data sanitization (Lison et al., 2021), the application of differential privacy algorithms(Abadi et al., 2016; Anil et al., 2021; Li et al., 2021; Tramèr et al., 2022; Basu et al., 2021), data deduplication (Kandpal et al., 2022), and knowledge unlearning (Jang et al., 2022). These techniques aim to prevent the generation of memorized content. However, they also come with certain drawbacks. Data sanitization

assumes that private information can be easily identified and is not context-dependent. Differential privacy can lead to lower-quality generative models (Anil et al., 2021). On the other hand, knowledge unlearning restricts the number of samples that can be forgotten at once to avoid degrading the overall capability of the language model, which may limit its effectiveness in real-world scenarios.

In this study, we propose DeMemorization (DeMem), a reward-based (un)learning framework for language models. DeMem leverages a paraphrasing policy to address memorization, using a negative similarity metric as a reward to encourage the language model (LM) to unlearn.

Given samples of prefixes and suffixes from the original pre-training data of the language model, we use a prefix as input for the language model to generate the suffix; then, we compute the negative BERTScore (Zhang et al., 2019) to measure the dissimilarity between the true suffix and generated suffix, the dissimilarity scores are then regarded as a reward signal to maximize in the training process, which guarantees that the approximate memorization will be mitigated.

For instance, given a training sample like "Alice Green lives at 187 Bob Street," where the prefix is **"Alice Green lives at"** and the suffix is **"187 Bob Street"**, our goal is to have the fine-tuned LM paraphrase the suffix as **"12 Red Street."** This paraphrasing approach minimizes the memorization relationship between the prefix and suffix without erasing the training sample from the LM's parameters or replacing it with meaningless content, which can negatively impact the LM's performance

We conducted experiments using GPT-Neo and OPT LMs (with models ranging from 125M to 2.7B parameters) (Black et al., 2021; Zhang et al., 2022). DeMem achieved little to no performance degradation on the initial LM capabilities measured via nine common NLP classification benchmarks (Hellaswag (Zellers et al., 2019), Lambda (Paperno et al., 2016), Winogrande (Sakaguchi et al., 2021), COPA (Roemmele et al., 2011), ARC-Easy, ARC-Challenge (Clark et al., 2018), Piqa(Bisk et al., 2020), MathQA (Amini et al., 2019), and PubmedQA (Jin et al., 2019)).

We also evaluate DeMem on increasing the context of the prefix, as many studies show that as a longer context is provided, the memorization ratio increases (Carlini et al., 2021, 2022). The proposed framework makes no explicit, implicit as-

sumptions or limitations about the data's structure or size to be protected. Also, unlike the DP methods, the proposed framework does not apply any partition mechanism to split the data into public data and private data; as language data cannot be partitioned(Brown et al., 2022), we apply the policy on all training data as defining, partitioning data into private and public, and limiting the number of samples inadequate in the real-world scenarios.

To summarize, our main findings are the following:

- Using a reinforcement learning feedback approach results in little to no performance degradation of general capabilities while being practical, consistent, and independent of increasing the number of protected samples. At the same time, maintaining the fluency and coherence of the generated samples.

- As the language model size increases, the convergence rate improves. Convergence refers to the model-generated suffixes diverging significantly from the original ones while the perplexity difference between generated and original examples decreases.

- As the size of a language model increases, the dissimilarity score increases. This suggests that larger models may tend to "forget" the memorized data faster.

- Combining Deduplication with DeMemorization enhances privacy with insignificant degradation(∼0.5%) in the Language model performance.

## 2 Background

### 2.1 Memorization Definitions

In the context of memorization in large language models, we follow the definition proposed by (Lee et al., 2021), which introduced approximate memorization. Given a string S, splitted into prefix (P) and suffix $(S_T)$. We fed the prefix to the LM to get the generated suffix $(S_G)$. The memorization is measured with the chosen edit distance between the true and generated suffix. In our study, we choose the edit distance to be a similarity measure (SacreBLEU (Post, 2018)) as proposed in (Ippolito et al., 2022), to be able to capture the approximate memorization, not just the "Eidetic memorization" (Carlini et al., 2021) as the definition of verbatim

memorization fails to include more subtle forms of memorization (Ippolito et al., 2022).

## 2.2 RL In Language Models

Unlearning undesirable behaviors is more compatible with the reinforcement learning (RL) paradigm. In the realm of NLP, RL has been employed to enhance scalar metrics through reward optimization (Ramamurthy et al., 2022; Ziegler et al., 2019; Ouyang et al., 2022). Lately, RL has gained prominence for addressing undesirable behavior, including toxicity, social biases, and offensive speech. This is accomplished by using Proximal Policy Optimization (PPO) (Schulman et al., 2017) to optimize a Language Model (LLM) based on a reward model. In this paper, we investigate using RL with a language model to mitigate privacy risks associated with memorization.

## 3 Related Work

In this section, we delve into recent studies to mitigate memorization in language models, which can be categorized into three main approaches: data pre/post-processing, differential privacy methods, and knowledge unlearning.

**Data Pre/Post-Processing:** This approach reduces memorization in training data by applying filters before or after feeding it into the language model. One method is data deduplication (Kandpal et al., 2022), which removes duplicates and improves model performance. However, it only partially protects against memorization as the model can still memorize non-duplicate sequences. Another approach is "MemFREE decoding" (Ippolito et al., 2022), which efficiently checks the memorization in the LM generation by an n-gram in the training dataset.

**Differential Privacy (DP):** is a widely-used technique for training models to prevent memorization of individual training examples (Abadi et al., 2016). While effective for fine-tuning language models (Yu et al., 2021; Li et al., 2021), DP often reduces performance compared to non-private models (Anil et al., 2021). State-of-the-art language models are typically trained without DP, using large amounts of data and computational resources. DP algorithms are computationally expensive, slower to converge, and have lower utility compared to non-private methods (Anil et al., 2021). Applying DP to language data is challenging due to defining private information

boundaries (Brown et al., 2022).

**Knowledge Unlearning (UL):** is an effective method that reverses the training objective of minimizing the negative log-likelihood for forgotten tokens. It minimally affects language modeling performance in larger models for a small number of samples. UL has two approaches: batch unlearning for multiple samples and sequential unlearning for smaller chunks. However, unlearning a large number of samples at once significantly degrades average language model performance. While UL effectively addresses memorization, it has not been tested on sample sizes larger than 128. Also, It does not preserve fluency or coherency for generated suffixes, which are crucial for practical applications.

**In this work,** we compare our proposed method with a data-preprocessing approach proposed by (Kandpal et al., 2022), which shows that deduplicating helps minimize data memorization. While this method is effective, we show that memorization is still high in the LMs pre-trained with this approach; thus, we show that combining pre-processing with our approach, "DeMemorization," effectively mitigates memorization. We also compare our method with UL and show it is not inadequate or impractical in real-world scenarios due to a limited number of samples to forget at once.

## 4 Methodology

### 4.1 DeMemorization Via Dissimilarity Policy

DeMemorization framework operates by learning a paraphrasing policy to mitigate memorization risks. We divide each sample into prefixes and suffixes using an LM and a subset of pre-training data. The unlearning process is as follows: we select a prefix P and a true suffix $S_T$, then input the prefix into the pre-trained LM to produce a suffix $S_G$. Using a negative similarity metric, we evaluate how the generated suffix is dissimilar to true. We use that as a reward signal to encourage the LM to develop a paraphrasing policy, generating dissimilar tokens to minimize memorization. These steps can be summarized as follows:

$$P, S_T \sim D_t \qquad (1)$$
$$S_G = f_\theta(s_{G_{i+1}}|x_{P_1}, ..., x_{P_i}) \qquad (2)$$
$$Dis_{Score} = -BERTScore(S_G, S_T) \qquad (3)$$

### 4.1.1 Reward Function

To yield the desired outcome of paraphrasing to mitigate memorization risk, we need to employ a similarity function to achieve this goal. The proposed reward function should allow changes in words or even the entire sentence while preserving the semantic meaning. Also, while learning the paraphrasing technique, we aim to ensure that the fine-tuned or Dememorized LM stays within the original LM to avoid potentially less coherent and relevant generation.

**Learning Dissimilarity with BERTScore.** To achieve the dissimilarity goal, we employ BERTScore. One advantage of BERTScore over other contextual embedding methods is the ability to operate on pairwise tokens using contextual embeddings, providing a more flexible definition of dissimilarity in our context. This flexibility means that BERTScore can yield a high similarity score for different words that share the same entity, encouraging the language model to learn a paraphrasing policy effectively. We employed the F-score metric produced using BERTScore.

**Achieving Stability Via KL Penalty.** To achieve the stability goal, we introduce a KL divergence penalty term to quantify the dissimilarity between these two policies. This step helps ensure that our optimization process remains within a trustworthy region. The KL divergence, calculated for the policies, is expressed as:

$$KL(\theta||\theta_c) = \sum_{i \in [1,t]} \pi_\theta(a_i|s_i) \cdot \log \frac{\pi_\theta(a_i|s_i)}{\pi_{\theta_c}(a_i|s_i)} \quad (4)$$

Here, we denote $\theta$ as the pre-trained policy, representing a model that has undergone initial training without fine-tuning. Additionally, we introduce $\theta_c$ as the updated policy, which signifies the policy after fine-tuning or further training.". We deduct KL divergence with default value weight $\beta = 0.2$ as a penalty term.

### 4.1.2 Policy Optimization Via PPO

To optimize the policy, we employ a Proximal Policy Optimization (PPO) methodology, incorporating a top-p sampling rate of 0.95, a technique commonly referred to as Natural Language Policy Optimization (NLPO), as elaborated in-depth in (Ramamurthy et al., 2022) (please refer to Appendix A for comprehensive elucidation). A value network $V$ is included beside the language modeling head to estimate the value function. The batch size is 32 for all models; we selected a specific number of steps for each model as the convergence rate for each model is different. We mean by convergence in this context that the model-generated suffixes become significantly different from the original suffixes but without a considerable loss in the perplexity as the difference between the perplexity of the generated examples and original examples becomes smaller, so we selected the appropriate number of steps that balance between these goals.

## 4.2 Measuring Memorization In Language Models

As mentioned in subsection 2.1, we adopt the concept of approximate memorization, as it provides a more precise and adaptable approach to capturing subtle forms of memorization compared to the limitations of exact memorization. We employ a widely accepted text similarity measure from standard Natural Language Processing (NLP) evaluation techniques to quantify approximate memorization accurately: the **SacreBLEU metric**. SacreBLEU is an improved version of BLEU, known for its stability in measuring the quality of machine-generated text.

To measure forgetting, we consider the negative of SacreBLEU. By utilizing SacreBLEU as a metric for estimating approximate memorization, we define DeMemorization or forgetting as the process of minimizing the relationship between the given prefix $P$ and the suffix $S$.

This relationship represents the information that the adversary seeks to extract based on the given prefix. The metric we mentioned quantifies this relationship. In an example scenario, an adversary has the personal email address **"bob@adam.com"** and seeks to obtain the password. If the LM has memorized this association, it can provide **the password "12345"** when given the email, however, by minimizing or altering their relationship. LM can generate a different suffix as **the password "0912,"**,

As a result, the generated suffixes are valid and meaningful output without memorizing sensitive information. This approach achieves the dual objectives of preserving the LM's general capability and the fluency of generated suffixes while ensuring privacy. Also, the solution is more practical in real-world situations than completely removing all information, which can negatively impact the capabilities of the language model (LM).

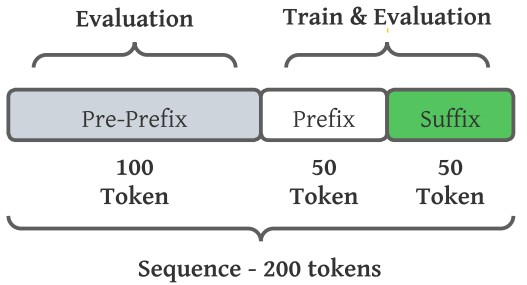

Figure 2: Illustration of sequence splitting in the training & evaluation data.

# 5 Experiments

In this section, we begin by introducing the dataset used for training and assessing the paraphrasing policy. Subsequently, we assess the overall performance of the dememorized LM general performance on nine benchmarks. We then establish the baseline methods for comparison. Finally, we define the evaluation metrics that enable us to measure the memorization and the performance in downstream tasks.

## 5.1 Experimental Settings

### 5.1.1 Memorization Dataset

We employed a subset of the Pile dataset, released as a benchmark for training data extraction attacks on large Language Models. Generally, the Pile dataset contains data from 16 different sources (e.g., books, Web scrapes, open source code). We used this version of the subset [1], designed to be easy to extract to assess targeted attack performance. The dataset contains only 15,000 samples since the full version has not been released yet. Each sample consists of 200 tokens sampled randomly from the Pile training set. The topics included in the subset are code, news, logs, conversations, copyrights, links, etc. Most of them are in the English language. The dataset is splitted into 13,500 samples for training and 1,500 samples for testing.

**Training & Evaluation Data.** Each sample consists of a 200-token sequence divided into 100 pre-prefix tokens, 50 prefix tokens, and 50 suffix tokens. During the training phase, we exclusively utilized the prefix and suffix tokens. However, we tested the model in two different settings during the

---

[1] https://github.com/google-research/lm-extraction-benchmark

evaluation phase. In the first setting, we evaluated the model's ability to predict the suffix when provided with only the prefix. In the second setting, we evaluated the model's capability to predict the suffix when given the pre-prefix and prefix. This evaluation assessed the model's capacity to protect against acquiring additional information or knowledge. A longer context in a language model can be considered a form of attack (Carlini et al., 2022). The sequence splitting is illustrated in Figure 2.

### 5.1.2 Downstream Tasks

To ensure stronger privacy protections for language models (LMs) without compromising their original capabilities, we undertake a comprehensive evaluation that encompasses both privacy risks and the inherent strengths of LMs. This evaluation involves quantifying the LMs' performance across various classification tasks to assess their general capabilities. The tasks include Hellaswag (Zellers et al., 2019) and Lambada (Paperno et al., 2016) benchmarks, which gauge linguistic reasoning abilities, as well as Winogrande (Sakaguchi et al., 2021) and COPA (Roemmele et al., 2011), which measure commonsense reasoning abilities. Additionally, we utilize ARC-Easy, ARC-Challenge (Clark et al., 2018), Piqa (Bisk et al., 2020), MathQA (Amini et al., 2019), and PubmedQA (Jin et al., 2019) benchmarks to assess scientific reasoning abilities. In addition to these classification tasks. We also measure the perplexity on the Wikitext (Merity et al., 2016) and Lambada (Paperno et al., 2016) datasets to gain insights into the LMs' language understanding and modeling. Whenever possible, we use the test sets for these evaluations; otherwise, we resort to the validation sets. Also, we did not report Lambada's perplexity & and accuracy as it shows high values for perplexity & low values for accuracy for the UL baseline. To discard the anomaly and better assess the performance, we report it in Appendix E.

### 5.1.3 Baseline Methods

Our experiments used the GPT-NEO family (125M, 1.3B, 2.7B), pre-trained on the publicly available 825GB Pile dataset. Additionally, we employed the OPT family (125M, 1.3B, 2.7B) (Zhang et al., 2022), which was pre-trained on a subset of the deduplicated version of the Pile, along with other corpora from diverse domains. OPT served as our baseline method for deduplication, as per (Jang et al., 2022), since the deduplicated version of GPT-

| Model | #Samples | N-SacreBLEU↑ | LM (ACC)↑ | LM (PPL)↓ | GEN (PPL)↓ | Epochs/Steps |
|---|---|---|---|---|---|---|
| NEO$_{125M}$ | 32 | 58.44 | | | 3.46 | - |
| | 128 | 58.41 | 43.36 | 32.28 | 3.83 | - |
| | 256 | 58.82 | | | 3.79 | - |
| +UL | 32 | 99.19 | 38.62 | 31098.06 | 19.77 | 18 |
| | 128 | 99.69 | 36.87 | 9683877.08 | 6.54 | 18 |
| | 256 | 99.63 | 36.34 | 25146.84 | 6.03 | 18 |
| +DeMem | 32 | 67.07 | | | 3.74 | |
| | 128 | 66.21 | 43.46 | 33.13 | 3.93 | 4 |
| | 256 | 67.05 | | | 3.95 | |
| NEO$_{1.3B}$ | 32 | 30.76 | | | 2.02 | - |
| | 128 | 34.7 | 48.93 | 16.16 | 2.18 | - |
| | 256 | 33.95 | | | 2.18 | - |
| +UL | 32 | 99.57 | 48.61 | 24.38 | 4.37 | 14 |
| | 128 | 98.33 | 41.55 | 188.65 | 5.83 | 8 |
| | 256 | 99.15 | 41.34 | 62.34 | 5.37 | 7 |
| +DeMem | 32 | 52.03 | | | 2.44 | |
| | 128 | 51.34 | 49.40 | 16.70 | 2.62 | 2 |
| | 256 | 52.58 | | | 2.65 | |
| NEO$_{2.7B}$ | 32 | 26.26 | | | 1.8 | - |
| | 128 | 27.25 | 52.67 | 13.93 | 1.92 | - |
| | 256 | 27.37 | | | 1.92 | - |
| +UL | 32 | 99.54 | 49.70 | 324.68 | 4.93 | 11 |
| | 128 | 97.77 | 47.42 | 41.50 | 9.67 | 8 |
| | 256 | 99.37 | 39.80 | 118.68 | 4.53 | 8 |
| +DeMem | 32 | 49.24 | | | 2.3 | |
| | 128 | 50.81 | 52.48 | 14.15 | 2.38 | 2 |
| | 256 | 50.91 | | | 2.35 | |

Table 1: Main Results: GPT-NEO averaged 5 random samples (s = 32, 128, and 256) for UL. NEO = initial GPT-NEO LM. UL+ = knowledge unlearning, DeMEM = DeMemorization. LM ACC. = average accuracy of 8 classification datasets, LM PPL = perplexity of Wikitext dataset, GEN PPL = perplexity of generated suffix. Steps for DeMEM & Epochs for UL

| Model | #Samples | N-SacreBLEU↑ | LM (ACC)↑ | LM (PPL)↓ | GEN (PPL)↓ | Epochs/Steps |
|---|---|---|---|---|---|---|
| OPT$_{125M}$ | 32 | 89.24 | | | 9.69 | - |
| | 128 | 90.98 | 41.28 | 31.94 | 9.76 | - |
| | 256 | 91.03 | | | 9.67 | - |
| +UL | 32 | 99.23 | 37.06 | 449131.90 | 12.16 | 9 |
| | 128 | 99.35 | 36.48 | 54917065.46 | 10.44 | 9 |
| | 256 | 99.21 | 37.19 | 114952.53 | 13.64 | 9 |
| +DeMem | 32 | 94.88 | | | 10.86 | |
| | 128 | 95.30 | 42.25 | 33.13 | 10.78 | 4 |
| | 256 | 95.61 | | | 10.58 | |
| OPT$_{1.3B}$ | 32 | 71.63 | | | 6.72 | - |
| | 128 | 71.96 | 51.65 | 16.41 | 6.92 | - |
| | 256 | 71.7 | | | 6.80 | - |
| +UL | 32 | 99.50 | 39.16 | ★ | 11.19 | 7 |
| | 128 | 99.84 | 38.67 | ★ | 7.93 | 8 |
| | 256 | 99.52 | 36.85 | ★ | 10.7 | 7 |
| +DeMem | 32 | 92.51 | | | 9.78 | |
| | 128 | 91.56 | 51.40 | 17.39 | 9.47 | 2 |
| | 256 | 91.91 | | | 9.25 | |
| OPT$_{2.7B}$ | 32 | 71.80 | | | 6.27 | - |
| | 128 | 67.56 | 53.74 | 14.31 | 6.48 | - |
| | 256 | 66.32 | | | 6.3 | - |
| +UL | 32 | 99.15 | 38.60 | ★ | 7.15 | 11 |
| | 128 | 97.87 | 41.06 | ★ | 13.43 | 7 |
| | 256 | 99.48 | 38.20 | ★ | 7.6 | 8 |
| +DeMem | 32 | 94.53 | | | 8.28 | |
| | 128 | 93.08 | 52.20 | 15.25 | 8.31 | 2 |
| | 256 | 93.24 | | | 8.16 | |

Table 2: Main Results: OPT averaged 5 random samples (s = 32, 128, and 256) for UL. UL = knowledge unlearning, DeMEM = DeMemorization. LM ACC = average accuracy of 8 classification datasets, LM PPL = perplexity of Wikitext dataset, GEN PPL = perplexity of generated suffix. ★ means that the value is so high, Reaching infinity. Epochs for UL & Steps for DeMeM.

NEO LMs by (Kandpal et al., 2022) were not publicly accessible. We also applied DeMemorization to the OPT LMs, which can be seen as a combination of the deduplication approach and DeMemorization, resulting in a significant enhancement in the privacy of these models. Furthermore, we included UL (Jang et al., 2022) as a second baseline method to highlight weaknesses and distinctions.

### 5.1.4 Implementation Details

For training, we utilized the training subset and fine-tuned the GPT-Neo & OPT LMs fine-tuned them for multiple iterations depending on the model size. To compare our proposed method with UL & deduplication, we followed the configuration proposed by (Jang et al., 2022) to ensure an adequate comparison, as we randomly sample s samples from the test subset and evaluate the models on those samples for UL since it forgets s samples only at once, we make the LM forget the s samples and then evaluated. To follow the same configuration, we show the average results of 5 random samplings of s samples for all of our experimental settings.

To explore the impact of increasing the sample size to be forgotten, we performed five random samplings of 32, 128, and 256. DeMemorization was carried out using a batch size of 32, and a default

value of learning rate of $1.41 \times 10^{-5}$ was applied to all models. We use the default value of KL Beta of 0.2 and a clip range of 0.2. The GPT-Neo & OPT LMs were employed using the official release in the Hugging Face library. For UL training and memorization evaluation, we utilized the official code provided by the authors. For the selection of hyperparameters, see Appendix F. In downstream tasks, we employed the lm-evaluation-harness framework (Gao et al., 2021) for all baseline methods.

### 5.1.5 Evaluation Metrics

We conducted a comprehensive evaluation of DeMemorization and baseline methods, employing a multi-perspective approach to assess their effectiveness in three key areas:

**(1) Measuring Forgetting:** As mentioned in subsection 4.2, we employed negative Sacre-BLEU to quantify memorization.

**(2) Evaluating Generated Suffixes:** To assess text fluency, we utilized the perplexity score of the underlying original model before forgetting. This metric enabled us to assess the grammatical correctness and coherence of the generated suffixes.

**(3) Performance on Downstream Tasks:** We assessed the performance of the unlearned models across nine classification tasks, employing accuracy scores and perplexity measurements on Wiki-

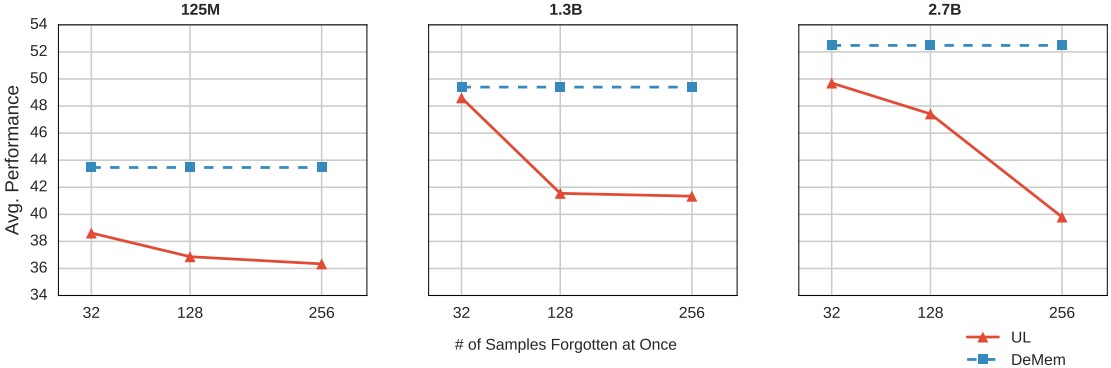

Figure 3: Average LM performance on the 8 benchmarks when varying the total number of samples forgotten for NEO (125M, 1.3B, 2.7B).

text and Lambada.

## 5.2 Experimental Results & Discussion

We conducted comprehensive experiments to assess the performance of DeMemorization against the baseline methods. Our main observations are as follows:

### 5.2.1 Overview of The DeMemorization Performance

We comprehensively evaluated the DeMemorization approach on nine classification tasks, wikitext for perplexity, and the generated samples. The evaluation results, as shown in Table 1, demonstrate that the DeMemorization approach effectively provides privacy and decreases the memorization for GPT-NEO while maintaining the LM general capability, measured by evaluating the classification tasks. It also maintains the fluency of the general LM and generated suffixes. On the other hand, the UL approach provides more robust protection since it removes the data points completely from the training data, which lowers the general LM capability by a large margin. This is effective privacy-wise but needs to be more practical from the performance perspective. Thus, we tried to balance this tradeoff by employing the DeMemorization approach. We provide the results for each dataset in Appendix E for reference.

### 5.2.2 Deduplication With DeMemorization & UL

We included OPT LMs as a baseline for the preprocessing technique, which applies deduplication to decrease memorization. Deduplicating the training data has effectively mitigated memorization, as Table 1, Table 2 demonstrate. OPT models (deduplicated) exhibit higher N-sacreBLEU scores than NEO (non-duplicate version) models while achieving similar or better performance in downstream tasks. However, even in these models, memorization remains high, as only a portion of the memorized samples are duplicates.

Therefore, we explored the UL approach and DeMemorization. The models that utilized both frameworks benefited significantly and became more robust privacy LMs. While UL reduced memorization by approximately 99% of N-sacreBLEU, it also negatively impacted the general capability of the LM, resulting in an ~11% difference from the original LM across various configurations. On the other hand, DeMemorization achieved comparable results to UL, with a reduction of ~94% in memorization, without the need to completely remove training data points from the LM parameters. In comparison, the loss in general LM capability was insignificant, at around ~0.5%, in the case of 125M and NEO 1.3B DeMemorization, even enhanced performance. These findings suggest that employing a combination of deduplication and DeMemorization effectively mitigates memorization while maintaining the general capability of the LM. Since data deduplication is applied in most of the recent & large language models (Penedo et al., 2023; Touvron et al., 2023; Biderman et al., 2023; Taylor et al., 2022; Scao et al., 2022; Black et al., 2022), we believe our approach combined with deduplication will effectively mitigate memorization.

### 5.2.3 Number of Samples, Stability, & Universal Policy

We investigated the impact of increasing the number of samples on the performance of both UL and DeMemorization. In line with the findings from

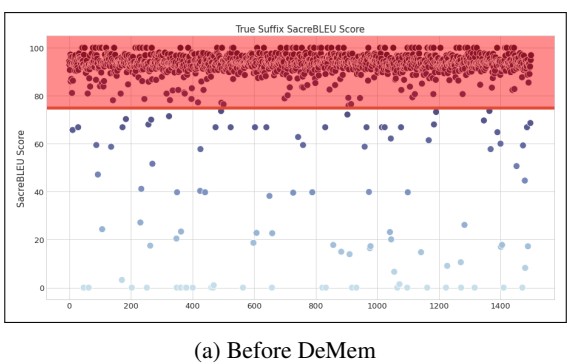
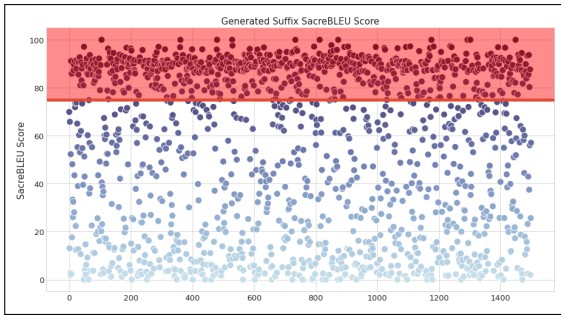

(a) Before DeMem               (b) After DeMem

Figure 4: Threshold of 75% SacreBLEU of The Generated Samples Before & After DeMemorization For Neo 2.7B Longer Context.

(Jang et al., 2022), UL is sensitive to the number of samples being unlearned simultaneously. Our experimental results validate this observation in Table 1, Table 2. As the number of samples increases, we observe a decrease in the LM's performance. On the other hand, DeMemorization demonstrates a different behavior as it is unaffected by the number of samples as shown in Figure 3. In DeMemorization, the LM is fine-tuned one-time using negative similarity as a reward during training, followed by evaluation on a separate test set. This allows the model to learn a universal policy to forget an unlimited number of samples. Here, the term *"unlimited"* signifies the absence of any restrictions, assumptions, or re-training of the LM regarding the number of samples to be unlearned.

In UL, however, the model is fine-tuned and evaluated on the same samples to forget them at a time. To unlearn or forget multiple samples, the model needs to undergo fine-tuning multiple times through sequential or batch unlearning. In each iteration, the model is fine-tuned with a specific number of samples (typically 32, as suggested by the authors) to prevent a decrease in the LM's overall capability. This can be regarded as an assumption about the number of samples to be protected simultaneously, leading to an incomplete solution. See Appendix G to highlight more UL framework assumptions.

### 5.2.4 Perplexity of WikiText & Generated Suffix

Perplexity serves as a crucial metric for assessing the overall performance of a Language Model (LM) in terms of its ability to generate fluent and coherent text. We computed perplexity for Wikitext and presented the results in Table 1, Table 2.

DeMemorization had a minimal impact on per-

| Model | #Parameters | BEFORE | | AFTER | |
|---|---|---|---|---|---|
| | | N − SacreBLEU ↑ | PPL ↓ | N − SacreBLEU ↑ | PPL ↓ |
| NEO | 125M | 45.74 | 4.12 | 55.04 | 4.15 |
| | 1.3B | 59.58 | 6.64 | 88.91 | 7.68 |
| | 2.7B | 10.55 | 1.41 | 32.66 | 1.54 |
| OPT | 125M | 89.35 | 11.99 | 94.47 | 12.38 |
| | 1.3B | 59.58 | 6.64 | 88.91 | 7.68 |
| | 2.7B | 56.35 | 5.95 | 89.37 | 6.76 |

Table 3: Comparsion of Negative SacreBLEU & Perplexity Means Before & After Applying The Framework On a Longer Context; 100 Extra Tokens Combined With The Prefix

plexity for all models. UL showed significantly higher perplexity in some cases, even reaching infinity. UL's high perplexity is attributed to its gradient ascent approach, which softens the probability distribution and leads to a more uniform distribution and higher perplexity. However, this softening procedure degrades LM performance as the model becomes less confident in generating tokens. We also evaluated the perplexity of unlearned samples, which is crucial in practical applications where the unlearned data domain is used. DeMemorization caused an average degradation of approximately 0.5% in NEO models and around 1.5% in OPT models. UL exhibited higher degradation in both models due to the complete removal of corresponding data points from the model parameters.

### 5.2.5 Protection Against Discoverability Phenomenon

Discoverability phenomenon refers to the observation that some memorization only becomes apparent under certain conditions, such as when a model is prompted with a sufficiently long context. (Carlini et al., 2022) found that the fraction of extractable sequences increases in a log-linear fashion with the number of tokens in the context.

| Prefix | True Suffix | Generated Suffix-Before | Generated Suffix-After | N-SacreBLEU$_B$ | N-SacreBLEU$_A$ | PPL$_B$ | PPL$_A$ |
|---|---|---|---|---|---|---|---|
| "POT-Creation-Date: 2017-02-24 11:50-0800\n" "PO-Revision-Date: YEAR-MO-DA HO:MI+ZONE\n" "Last | -Translator: FULL NAME <EMAIL@ADDRESS>\n" "Language-Team: LANGUAGE <█@█.org>\n" "Language: \n" "MIME-Version | -Translator: FULL NAME <EMAIL@ADDRESS>\n" "Language-Team: LANGUAGE <█@█.org>\n" "MIME-Version: 1.0\n" "Content | -Translator: FULL NAME <EMAIL@ADDRESS>\n" "Language-Team: English (India) (http://www.transifex.com /odoo/teams/2027/English/)\n" | 12.97 | 62.38 | 1.68 | 1.98 |
| the Free Software Foundation, either version 3 of the License, or * (at your option) any later version. * * weid-java | -sdk is distributed in the hope that it will be useful, *   but WITHOUT ANY WARRANTY; without even the implied warranty of *   MERCHANTABILITY or FITNESS | org is distributed in the hope that it will be useful, *   but WITHOUT ANY WARRANTY; without even the implied warranty of *   MERCHANTABILITY or FITNESS FOR A PARTIC | .com * * | 16.41 | 99.95 | 1.93 | 2.11 |
| █@█.com>, Da█ <D█@█.com>, R█ █u█ <█@█.com>, M█ M█ <mm█ | .com>, S█ M█ <S█M█ <█@█.com>, T█ B█ <█@█.com>, R█ █S█ <█@█ | .com>, S█ J M█ <SJM█ @enron.com>, S█ K█ <S█ J.K█ @█ .com>, A█ J█ .com>, S█ | .com>, S█ J█ <S█ J.M█ @█ .com,> cc: Subject: FERC MeetingThe FERC meeting is scheduled for Tuesday, June 2, 2001 at | 69.87 | 86.04 | 1.96 | 2.60 |
| in this distribution, be it the RC4, RSA, * lhash, DES, etc., code; not just the SSL code. The SSL documentation * included with this distribution is covered by the same copyright terms * except that the holder is Tim | Hudson (t█@cr█ .com). * * Copyright remains E█ Y█ , and as such any Copyright notices in * the code are not to be removed. * If this package is used in a product, E█ Y█ should be given | Hudson (t█@cr█ .com). * * * * * * * * * * * * * * * * * * * | Hudson. * * * * The SSL version of this file is 2.0.0.\n\n * \r\n * \r\n * \r\n * The ASN.1 notation for | 80.12 | 96.52 | 3.80 | 6.64 |

Figure 5: Generated & True Suffixes given the prefixes before & after applying DeMem. Green indicates that this part is memorized according to the true suffix, while red indicates that it's dissimilar.

For example, with a context of 50 tokens, approximately 33% of training sequences can be extracted from the NEO-6B model. However, with a context of 450 tokens, this percentage rises to 65%.

We evaluated our DeMemorization approach by increasing the prefix context from 50 to 150 tokens. The results in Table 1, Table 2 show that extending the context does not significantly impact the 125M model in NEO, with a forgetting rate decrease from 58.44% to 45.47%, and has no effect in OPT-125M. However, for larger models like 1.3B and 2.7B, a longer context considerably reduces the forgetting rate by approximately 49% in NEO and around 10% in OPT. Nevertheless, DeMemorization effectively counters this type of attack, increasing the forgetting rate by approximately 10% for the 125M model and approximately 30% for larger sizes in OPT & NEO as shown in Table 3. This demonstrates the universality and generalizability of the learned policy across various scenarios.

### 5.2.6 Approximate Memorization Threshold

Based on (Ippolito et al., 2022), a BLEU score of 75% for the generated suffix is considered a suitable threshold for determining approximate memorization. However, our investigation found that even a threshold as low as 50% after applying the framework can mitigate this issue. Nevertheless, we chose to use the widely accepted threshold of 75% to demonstrate the effectiveness of our framework. Applying DeMemorization to the LM resulted in a significant decrease in memorized samples. For GPT-Neo 1.3B and 2.7B, approximate

memorization examples decreased from 910 to 497 and 1036 to 321, respectively (refer to Appendix B for other models). The red region in Figure 4 represents samples with scores equal to or above 75%. After DeMemorization, the distribution of samples spreads more evenly across different values instead of being concentrated beyond the 75% threshold. Box plots (see Appendix D) confirm the efficiency of the DeMemorization approach, as evidenced by the median of the sample's distribution before and after DeMemorization.

### 5.2.7 Qualitative Results

Figure 5 demonstrates that the framework is capable of learning a policy that reduces or eliminates the amount of memorized personal data, such as email addresses. However, it should be noted that in certain instances, this can increase perplexity. More samples demonstrating Dememorization can be found in Appendix C.

## 6 Conclusion

In this paper, we present a novel framework that tackles the problem of training data memorization in LLMs. We achieve this by employing an RL paraphrasing policy. Through extensive evaluations conducted in diverse settings, we demonstrate the effectiveness of our approach. Our framework successfully reduces memorization by significantly decreasing the SacreBLEU score while preserving the overall capabilities of the LM as measured by nine classification benchmarks.

## Limitations

One of the limitations of our work is that it relies on a single scalar reward for optimization, as the problem has dual objectives: dissimilarity and perplexity. To overcome this limitation, we suggest exploring other techniques, such as Multi-objective Reinforcement Learning, which can potentially enhance performance and optimize both objectives simultaneously.

## Ethics Statement

Improving the large language model to be privacy-preserving is crucial since the language models have become more prominent and involved in many applications in multi-aspect of life. Ensuring the data privacy of those models is vital since some adversary may be able to reach that information. To make those models widely used, we have to guarantee they cannot emit private data. In this paper, we hope our work will serve as a foundation for developing new and innovative solutions to the problem of approximate memorization in large language models since verbatim memorization can give a false sense of privacy, as earlier work suggested. Our proposed framework provides a promising approach to addressing this issue. Further research and experimentation in this area can lead to even more effective methods for reducing memorization in these models. Our work also highlights the importance of considering both the computational cost and the performance trade-off when developing new techniques for addressing memorization in large language models.

## Acknowledgements

The authors would like to thank Niloofar Mireshghallah for helpful feedback.

This research is supported by the Vector Scholarship in Artificial Intelligence, provided through the Vector Institute and Natural Sciences and Engineering Research Council of Canada (NSERC) by NSERC Discovery Grant. This research was enabled in part by support provided by Compute Ontario and the Digital Research Alliance of Canada.

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

## A Natural Language Policy Optimization vs PPO

To tackle the challenge posed by large action spaces in language generation tasks, the NLPO (Natural Language Policy Optimization) framework was proposed. Previous research by (Ramamurthy et al., 2022) highlighted the difficulties faced by existing RL algorithms when dealing with models like GPT-2/3 and T5, which have extensive vocabularies of 50K and 32K tokens, respectively, and this issue becomes even more pronounced with newer models. NLPO introduces a masking policy that is periodically updated and incorporates a top-p sampling technique during training. This technique helps address the dilemma of balancing the inclusion of task-relevant information while mitigating the risk of reward hacking. By extending the PPO (Proximal Policy Optimization) algorithm, NLPO aims to enhance the stability and effectiveness of training language models. NLPO achieves this by employing top-p sampling through generating, which restricts the selection of tokens to a smaller setting where the cumulative probability surpasses a given threshold parameter, p (Holtzman et al., 2018).

## B Displaying Approximate Memorization Threshold

Recent studies suggested that approximate memorization occurs at the BLEU score of 75%; we follow this suggestion and demonstrate the effectiveness of the proposed framework in this section by comparing the number of samples that exceed this threshold before and after applying the framework.

$$\text{SacreBLEU}(\text{suffix}_G, \text{suffix}_T) > 0.75 \tag{5}$$

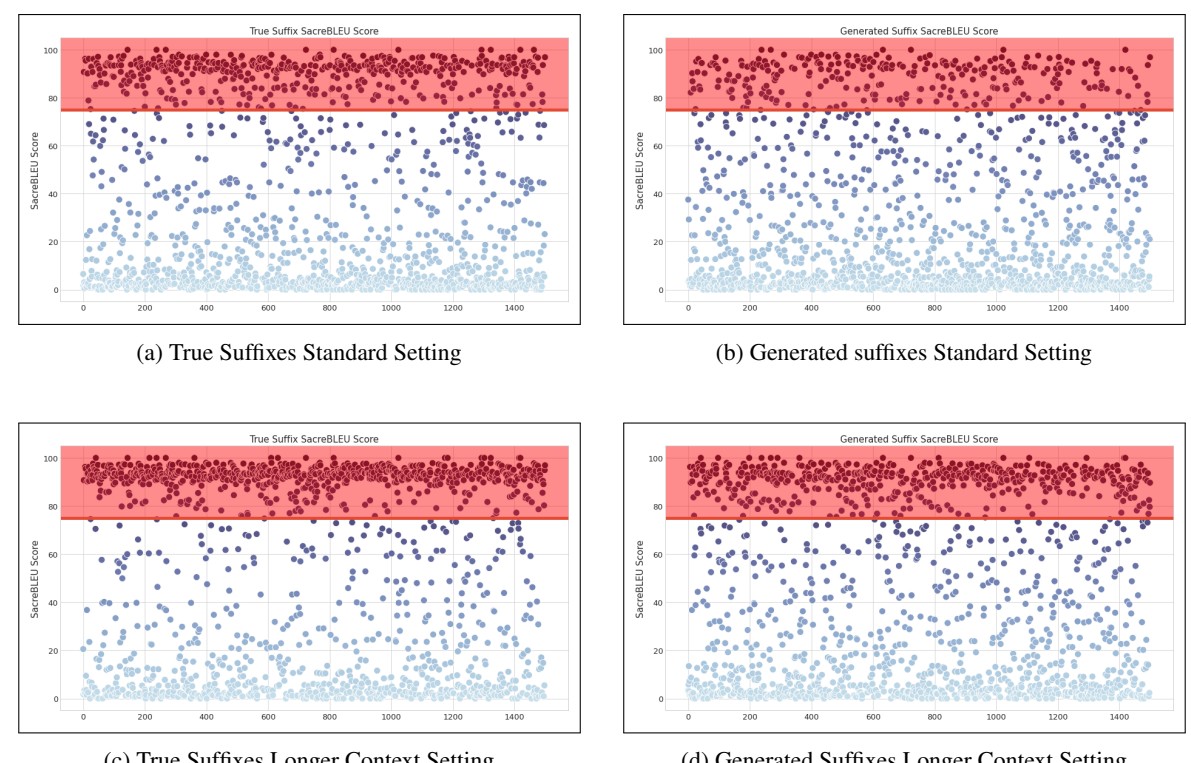

(a) True Suffixes Standard Setting      (b) Generated suffixes Standard Setting

(c) True Suffixes Longer Context Setting      (d) Generated Suffixes Longer Context Setting

Figure 6: Threshold of 75% Of The True & Generated Samples SacreBLEU For GPT-Neo 125M Standard Setting

As shown in Figure 6, the memorization ratio for the GPT-Neo 125M model is relatively low. However, when using standard and longer context settings, there are many instances where the samples are distributed on and beyond the 75% threshold. Despite this, after implementing the proposed framework, the distribution of samples is more evenly spread across various values rather than being concentrated solely in the region beyond the 75% threshold. In contrast to the other variation, GPT-Neo 1.3B & 2.7B have a large memorization ratio, especially in case of longer context; the framework effect can be seen obviously as many samples exceed the threshold in case of those variations as shown in Figure 7 and Figure 8.

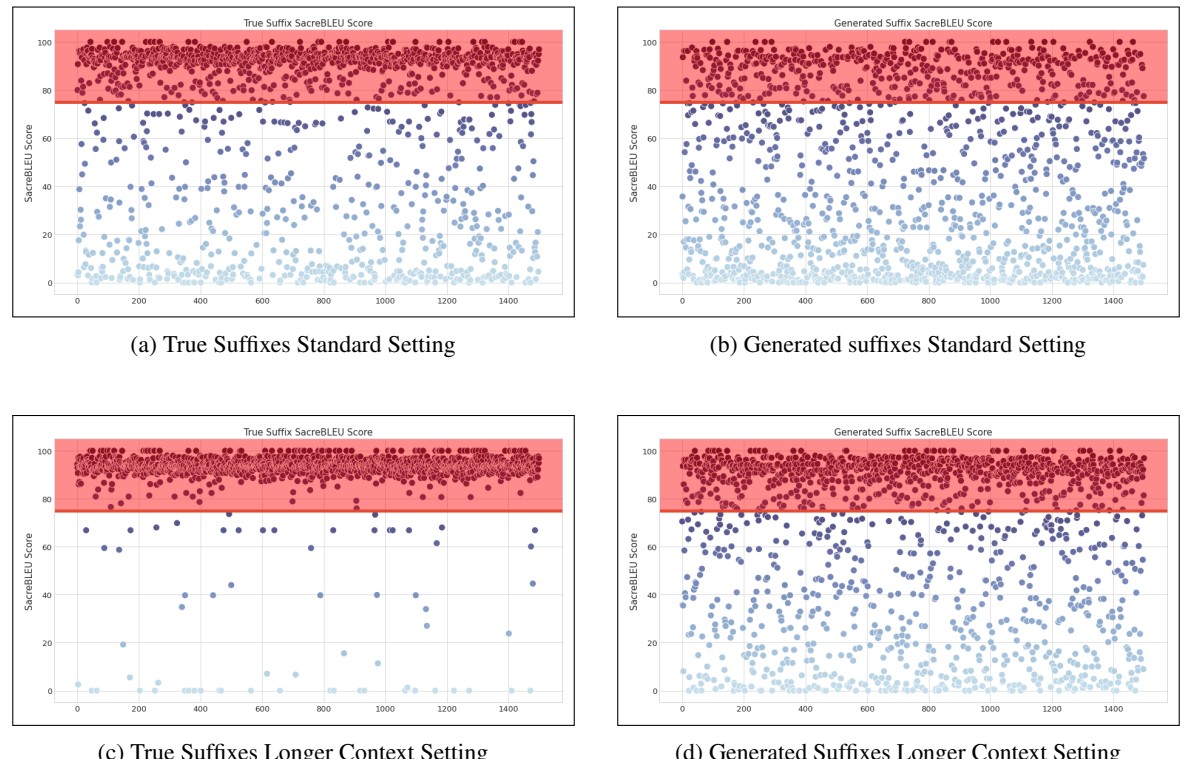

(a) True Suffixes Standard Setting

(b) Generated suffixes Standard Setting

(c) True Suffixes Longer Context Setting

(d) Generated Suffixes Longer Context Setting

Figure 7: Threshold of 75% Of The True & Generated Samples SacreBLEU For GPT-Neo 1.3B Standard Setting

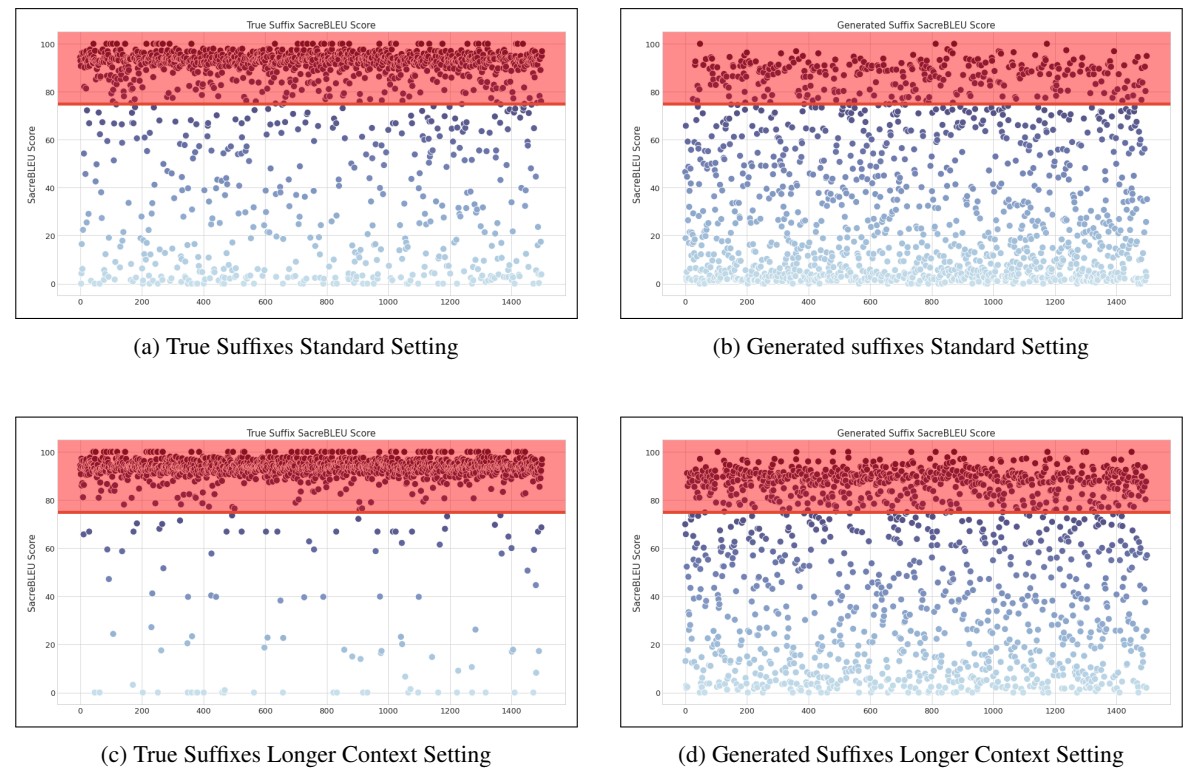

(a) True Suffixes Standard Setting

(b) Generated suffixes Standard Setting

(c) True Suffixes Longer Context Setting

(d) Generated Suffixes Longer Context Setting

Figure 8: Threshold of 75% Of The True & Generated Samples SacreBLEU For GPT-Neo 2.7B Standard Setting

# C   Qualitative Results

In this section, we demonstrate the effectiveness of our proposed framework by presenting a thorough analysis of samples generated before and after its application. To provide a comprehensive evaluation, we have chosen samples from various model sizes, including 125M, 1.3B, and 2.7B, and included examples from both standard and longer contexts. Additionally, we present samples from different training phases to showcase the learned policy's evolution over time. As previously mentioned, the policy initially focuses on replacing individual words or numbers to decrease the similarity between samples. As the training process progresses, the policy becomes more aggressive and replaces entire phrases, as shown in Figure 9.

| Prefix | True Suffix | Generated Suffix-Before | Generated Suffix-After | N-SacreBLEU$_B$ | N-SacreBLEU$_A$ | PPL$_B$ | PPL$_A$ |
|---|---|---|---|---|---|---|---|
| IF A STAY OF EXECUTION OF SENTENCE AND RELEASE UPON BAIL HAS BEEN PREVIOUSLY GRANTED BY THE TRIAL COURT OR THIS COURT, it is temporarily continued for a period not to exceed sixty days upon the bail previously posted. The purpose of a continued stay is to allow Appellant to file with the Supreme Court of Ohio an application for a stay during the pendency of proceedings in that court. If a stay is continued by this entry, it will terminate at the earlier of the expiration of the sixty day period, or the failure of the Appellant to file a notice of appeal with the Supreme Court of Ohio in the | forty-five day appeal period pursuant to Rule II, Sec. 2 of the Rules of Practice of the Supreme Court of Ohio. Additionally, if the Supreme Court of Ohio dismisses the appeal prior to expiration of sixty days, the | forty-five day appeal period pursuant to Rule II, Sec. 2 of the Rules of Practice of the Supreme Court of Ohio. Additionally, if the Supreme Court of Ohio dismisses the appeal prior to expiration of sixty days, the stay will terminate | case of a continuation. If the Supreme Court of Ohio denies the Appellant's motion for a stay, the stay will terminate at the earliest of the expiration of the sixty day period, or 7 days after the date of entry of this order, or | 6.74 | 87.6 | 1.60 | 1.97 |
| ModuleDirectories = true; loadScripts(document,'script');</script> <noscript> <div>JavaScript is disabled on your browser.</div> </noscript> <header role="banner" | "> <nav role="navigation"> <div class="fixedNav"> <!-- ========= START OF TOP NAVBAR ======= --> <div class="topNav"><a id="navbar.top"> <!-- | "> <nav role="navigation"> <div class="fixedNav"> <!-- ========= START OF TOP NAVBAR ======= --> <div class="topNav"><a id="navbar.top"><!-- --> | " class="ui-bar-nav-center d-none d-md-block d-lg-block"> </header> <div class="dropdown"> <button class="btn btn-primary dropdown-toggle" type="button" | 3.10 | 88.27 | 3.04 | 3.20 |
| . Related Posts Reader Interactions Submit Your CommentDisclaimer: Reviews on FitSmallBusiness.com are the product of independent research by our writers, researchers, and editorial team. User reviews and comments are contributions from independent users not affiliated with FitSmallBusiness.com's editorial team. Banks, issuers, credit card companies, and other product & service providers are not responsible for any content posted on FitSmallBusiness.com. As such, they do not endorse or guarantee any posted comments or reviews.Cancel reply You must be logged in to comment. Click the "Log in With LinkedIn" button below to connect instantly and comment. Social Contact Us Disclaimer: We spend hours researching and | writing our articles and strive to provide accurate, up-to-date content. However, our research is meant to aid your own, and we are not acting as licensed professionals. We recommend that you consult with your own lawyer, accountant, or other | writing our articles and strive to provide accurate, up-to-date content. However, our research is meant to aid your own, and we are not acting as licensed professionals. We recommend that you consult with your own lawyer, accountant, or other licensed professional for | writing our articles and strive to provide accurate, up-to-date content. However, our research is meant to aid your own personal understanding. We are not a professional body, and do not claim to be. This information is subject to change at any time. | 6.31 | 52.61 | 1.34 | 1.92 |
| * following conditions: * * The above copyright notice and this permission notice shall be included * in all copies or substantial portions of the Software. * * THE SOFTWARE IS PROVIDED "AS IS", WITHOUT WARRANTY OF ANY | KIND, EXPRESS * OR IMPLIED, INCLUDING BUT NOT LIMITED TO THE WARRANTIES OF * MERCHANTABILITY, FITNESS FOR A PARTICULAR PURPOSE AND NONINFRINGEMENT. IN * NO EVENT | KIND, EXPRESS\n * OR IMPLIED, INCLUDING BUT NOT LIMITED TO THE WARRANTIES OF\n * MERCHANTABILITY, FITNESS FOR A PARTICULAR PURPOSE AND NONINFRINGEMENT. IN * NO EVENT SHALL THE AUTH | KIND, * EXPRESS OR IMPLIED, INCLUDING BUT NOT LIMITED TO THE WARRANTIES OF * MERCHANTABILITY, FITNESS FOR A PARTICULAR PURPOSE AND NON-INFRINGED * AREA CONDITION. | 9.53 | 30.46 | 1.63 | 1.09 |

Figure 9: Suffixes that are memorized by the employed language models and the generated suffixes given the same prefix. Green indicates that this part is memorized according to the true suffix, while red indicates that it's dissimilar.

# D Median Comparison

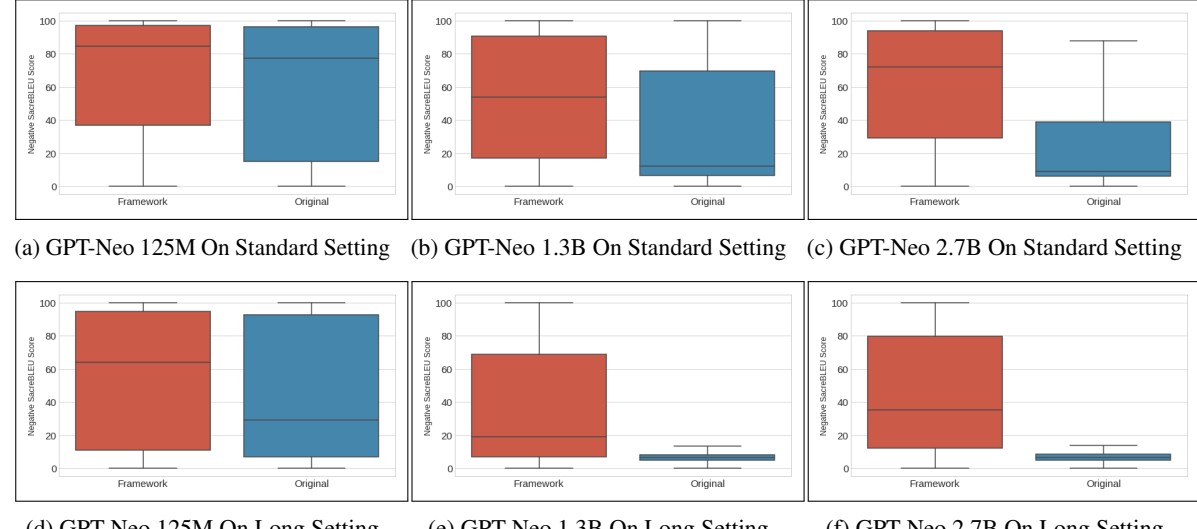

(a) GPT-Neo 125M On Standard Setting    (b) GPT-Neo 1.3B On Standard Setting    (c) GPT-Neo 2.7B On Standard Setting

(d) GPT-Neo 125M On Long Setting    (e) GPT-Neo 1.3B On Long Setting    (f) GPT-Neo 2.7B On Long Setting

Figure 10: Displaying The Negative SacreBLEU Distribution of The Models On Standard & Long Settings Before (blue) & After (orange) Applying The Framework

# E Results of Each Dataset

| Model | #Samples | Hella. (ACC) | Lamba. (ACC) | Wino. (ACC) | COPA (ACC) | ARC-E (ACC) | ARC-C (ACC) | Piqa (ACC) | MathQ (ACC) | PubQ (ACC) | Avg (ACC) |
|---|---|---|---|---|---|---|---|---|---|---|---|
| NEO$_{125M}$ | - | 28.66 | 37.35 | 50.43 | 64.0 | 43.72 | 19.11 | 63.05 | 22.78 | 55.10 | 43.36 |
| +DeMEM | - | 28.54 | 34.09 | 50.74 | 64.00 | 43.89 | 19.96 | 63.00 | 22.44 | 55.10 | 43.46 |
| +UL | 32 | 28.47 | 1.3 | 52.09 | 60.00 | 36.90 | 20.13 | 58.48 | 21.10 | 39.30 | 39.56 |
| | | 27.03 | 0.05 | 50.59 | 59.00 | 31.18 | 18.94 | 54.95 | 20.40 | 33.80 | 36.98 |
| | | 28.40 | 1.94 | 50.90 | 61.00 | 37.87 | 19.53 | 60.01 | 21.00 | 36.20 | 39.36 |
| | | 26.66 | 0.09 | 52.95 | 56.00 | 31.90 | 19.28 | 55.22 | 19.93 | 33.80 | 36.97 |
| | | 26.71 | 3.55 | 50.27 | 54.00 | 31.10 | 22.61 | 58.97 | 22.91 | 55.20 | 40.22 |
| +UL | 128 | 26.31 | 0 | 51.61 | 54.00 | 29.67 | 18.85 | 53.69 | 19.43 | 33.80 | 35.92 |
| | | 26.26 | 0.13 | 51.30 | 55.00 | 29.46 | 20.30 | 56.90 | 21.34 | 55.20 | 39.47 |
| | | 26.91 | 0.67 | 51.30 | 59.00 | 32.91 | 18.08 | 55.76 | 20.97 | 33.90 | 37.35 |
| | | 26.40 | 0 | 51.69 | 55.00 | 29.58 | 18.17 | 53.59 | .20.20 | 33.80 | 36.05 |
| | | 26.40 | 0 | 51.06 | 53.00 | 29.16 | 18.60 | 52.55 | 19.83 | 33.80 | 35.55 |
| +UL | 256 | 26.61 | 0 | 50.67 | 56.00 | 30.97 | 18.25 | 53.91 | 20.26 | 33.80 | 36.31 |
| | | 26.56 | 0 | 53.51 | 56.00 | 31.94 | 20.64 | 59.03 | 22.37 | 55.20 | 36.14 |
| | | 26.78 | 0.05 | 50.35 | 56.00 | 31.94 | 18.68 | 54.57 | 20.50 | 33.80 | 36.58 |
| | | 26.56 | 0 | 50.82 | 55.00 | 30.93 | 18.60 | 54.24 | 20.43 | 33.80 | 36.30 |
| | | 26.73 | 0 | 51.06 | 55.00 | 31.39 | 18.43 | 54.02 | 20.63 | 33.80 | 36.38 |
| NEO$_{1.3B}$ | - | 38.65 | 57.20 | 54.93 | 69.00 | 56.18 | 23.12 | 71.10 | 24.05 | 54.40 | 48.93 |
| +DeMEM | - | 38.73 | 51.71 | 55.48 | 73.00 | 55.17 | 23.63 | 70.72 | 23.65 | 54.80 | 49.40 |
| +UL | 32 | 39.00 | 24.23 | 54.69 | 74.00 | 54.25 | 25.25 | 69.58 | 23.11 | 52.00 | 48.98 |
| | | 38.50 | 32.56 | 54.61 | - | 72.00 | 55.47 | 25.25 | 69.47 | 23.31 | 50.80 |
| | | 35.51 | 65.70 | 53.82 | 75.00 | 51.13 | 22.86 | 68.11 | 24.22 | 55.00 | 48.21 |
| | | 38.34 | 61.01 | 54.69 | 69.00 | 53.66 | 23.80 | 69.26 | 24.69 | 53.60 | 48.38 |
| | | 37.41 | 64.48 | 56.66 | 73.00 | 52.94 | 23.63 | 69.15 | 23.85 | 53.70 | 48.79 |
| +UL | 128 | 27.96 | 8.81 | 52.88 | 55.00 | 30.00 | 19.28 | 56.03 | 21.57 | 54.80 | 39.69 |
| | | 33.32 | 59.88 | 57.30. | 66.00 | 47.34 | 21.58 | 65.72 | 25.19 | 55.10 | 46.44 |
| | | 26.89 | 0 | 52.09 | 54.00 | 27.98 | 20.30 | 53.21 | 20.33 | 34.40 | 36.15 |
| | | 30.11 | 40.52 | 53.90 | 65.00 | 42.29 | 20.81 | 61.91 | 23.68 | 55.00 | 44.09 |
| | | 31.14 | 6.63 | 55.72 | 63.00 | 42.29 | 19.53 | 63.05 | 22.17 | 34.00 | 41.36 |
| +UL | 256 | 28.79 | 5.93 | 52.17 | 55.00 | 33.45 | 18.85 | 56.80 | 21.60 | 56.70 | 40.42 |
| | | 29.43 | 39.53 | 53.35 | 60.00 | 36.57 | 19.28 | 59.63 | 23.24 | 55.20 | 42.09 |
| | | 28.02 | 28.10 | 53.90 | 62.00 | 34.97 | 19.11 | 58.75 | 23.31 | 54.60 | 41.83 |
| | | 29.19 | 31.34 | 52.40 | 55.00 | 35.47 | 19.36 | 58.75 | 22.61 | 55.20 | 41.00 |
| | | 29.92 | 15.11 | 52.88 | 56.00 | 37.07 | 19.11 | 58.86 | 21.94 | 55.30 | 41.38 |
| NEO$_{2.7B}$ | - | 42.71 | 62.24 | 57.70 | 79.00 | 61.06 | 27.47 | 72.19 | 24.05 | 58.30 | 52.67 |
| +DeMEM | - | 42.30 | 59.42 | 58.01 | 79.00 | 60.14 | 27.47 | 71.65 | 24.58 | 56.70 | 52.48 |
| +UL | 32 | 39.33 | 61.96 | 55.80 | 77.00 | 58.75 | 27.81 | 69.91 | 24.12 | 55.30 | 51.00 |
| | | 28.34 | 23.48 | 53.11 | 75.00 | 31.90 | 22.44 | 55.60 | 21.84 | 52.70 | 42.61 |
| | | 41.73 | 46.49 | 58.56 | 76.00 | 57.40 | 27.64 | 70.94 | 24.79 | 60.90 | 52.24 |
| | | 43.69 | 44.65 | 58.16 | 73.00 | 59.34 | 27.04 | 71.87 | 25.22 | 60.30 | 52.33 |
| | | 39.89 | 66.46 | 56.74 | 74.00 | 54.71 | 28.66 | 68.55 | 24.55 | 55.30 | 50.30 |
| +UL | 128 | 31.86 | 55.88 | 54.38 | 69.00 | 43.60 | 20.39 | 66.26 | 23.21 | 55.30 | 45.50 |
| | | 30.75 | 41.04 | 54.53 | 65.00 | 40.36 | 20.13 | 63.54 | 22.47 | 55.60 | 44.85 |
| | | 41.52 | 50.86 | 58.64 | 71.00 | 58.37 | 25.51 | 71.32 | 23.91 | 61.10 | 51.42 |
| | | 37.16 | 58.43 | 58.32 | 71.00 | 50.25 | 23.72 | 69.04 | 24.38 | 56.90 | 48.84 |
| | | 39.20 | 17.54 | 59.27 | 73.00 | 52.14 | 25.85 | 68.60 | 22.51 | 37.80 | 47.29 |
| +UL | 256 | 31.86 | 55.88 | 54.38 | 69.00 | 43.60 | 20.39 | 66.26 | 23.21 | 57.00 | 45.71 |
| | | 25.80 | 0.15 | 52.72 | 58.00 | 26.09 | 19.45 | 53.91 | 19.96 | 55.20 | 38.90 |
| | | 26.52 | 0.03 | 51.69 | 58.00 | 27.44 | 17.91 | 54.89 | 19.63 | 56.60 | 39.08 |
| | | 25.97 | 0 | 49.88 | 58.00 | 26.34 | 20.30 | 53.15 | 19.86 | 44.40 | 37.24 |
| | | 29.42 | 6.30 | 50.67 | 62.00 | 30.59 | 20.47 | 56.03 | 20.56 | 34.70 | 38.00 |

Table 4: Main Results: NEO averaged 5 random samples (s = 32, 128, and 256) for UL. UL = knowledge unlearning, DeMEM = DeMemorization. LM ACC = average accuracy of 8 classification datasets. Lambada Accuracy is excluded from the average due to anomalies

| Model | #Samples | Hella. (ACC) | Lamba. (ACC) | Wino. (ACC) | COPA (ACC) | ARC-E (ACC) | ARC-C (ACC) | Piqa (ACC) | MathQ (ACC) | PubQ (ACC) | Avg (ACC) |
|---|---|---|---|---|---|---|---|---|---|---|---|
| OPT$_{125M}$ | - | 29.21 | 37.92 | 50.28 | 66.00 | 43.52 | 19.11 | 63.00 | 22.04 | 37.10 | 41.28 |
| +DeMEM | - | 28.90 | 36.08 | 50.43 | 66.00 | 40.99 | 19.70 | 62.73 | 21.64 | 47.60 | 42.25 |
| +UL | 32 | 26.84 | 0.56 | 50.03 | 60.00 | 28.87 | 19.79 | 57.01 | 21.50 | 33.80 | 37.23 |
| | | 26.61 | 0.02 | 52.40 | 60.00 | 28.49 | 19.88 | 56.47 | 20.77 | 33.80 | 37.30 |
| | | 26.87 | 2.27 | 49.25 | 58.00 | 29.40 | 19.70 | 57.67 | 21.17 | 33.80 | 36.98 |
| | | 26.67 | 0.42 | 49.48 | 61.00 | 28.61 | 20.30 | 56.42 | 20.67 | 33.80 | 37.12 |
| | | 26.68 | 0.34 | 51.46 | 56.00 | 28.95 | 20.39 | 55.71 | 20.46 | 33.80 | 36.68 |
| +UL | 128 | 26.61 | 0.03 | 48.22 | 57.00 | 28.40 | 21.16 | 54.89 | 20.23 | 33.80 | 36.29 |
| | | 26.70 | 0.03 | 49.40 | 57.00 | 28.57 | 21.16 | 55.33 | 20.70 | 33.80 | 36.58 |
| | | 26.62 | 0 | 50.11 | 58.00 | 28.32 | 21.07 | 54.57 | 19.83 | 33.80 | 36.54 |
| | | 26.68 | 0.03 | 51.14 | 57.00 | 28.32 | 21.50 | 55.05 | 20.70 | 33.80 | 36.77 |
| | | 26.50 | 0.01 | 49.32 | 57.00 | 28.28 | 20.64 | 54.62 | 19.69 | 33.80 | 36.23 |
| +UL | 256 | 26.73 | 1.2 | 49.64 | 60.00 | 28.57 | 21.16 | 57.12 | 21.27 | 33.80 | 37.29 |
| | | 26.91 | 0.7 | 50.82 | 61.00 | 28.74 | 20.64 | 56.40 | 21.34 | 33.80 | 37.46 |
| | | 26.99 | 0.5 | 50.27 | 59.00 | 28.36 | 21.50 | 56.52 | 20.77 | 33.80 | 37.15 |
| | | 26.84 | 0.4 | 50.82 | 58.00 | 28.07 | 21.84 | 56.25 | 21.23 | 33.80 | 37.11 |
| | | 26.93 | 0.8 | 49.48 | 58.00 | 28.15 | 21.33 | 56.63 | 21.13 | 33.80 | 36.93 |
| OPT$_{1.3B}$ | - | 41.48 | 57.91 | 59.35 | 79.00 | 57.07 | 23.42 | 71.76 | 23.29 | 57.90 | 51.65 |
| +DeMEM | - | 41.57 | 53.74 | 60.45 | 78.00 | 55.13 | 24.91 | 70.83 | 23.85 | 56.50 | 51.40 |
| +UL | 32 | 30.37 | 0.64 | 52.17 | 59.00 | 28.15 | 23.72 | 56.69 | 20.50 | 55.50 | 40.76 |
| | | 30.60 | 4.48 | 51.30 | 59.00 | 29.71 | 23.29 | 57.61 | 20.93 | 40.40 | 39.10 |
| | | 30.07 | 0.81 | 51.93 | 60.00 | 27.73 | 23.03 | 56.47 | 20.90 | 51.20 | 40.16 |
| | | 28.13 | 0 | 51.77 | 54.00 | 27.18 | 22.35 | 55.27 | 19.83 | 33.80 | 36.54 |
| | | 30.28 | 1.88 | 50.82 | 63.00 | 29.04 | 22.18 | 57.07 | 21.23 | 40.40 | 39.25 |
| +UL | 128 | 27.10 | 0 | 51.14 | 53.00 | 24.07 | 21.75 | 55.93 | 19.26 | 55.20 | 40.76 |
| | | 27.49 | 0 | 51.38 | 51.00 | 24.32 | 22.26 | 56.20 | 19.29 | 55.20 | 39.10 |
| | | 27.34 | 0 | 50.90 | 54.00 | 24.66 | 23.20 | 55.76 | 19.09 | 55.20 | 40.16 |
| | | 28.53 | 0 | 51.06 | 60.00 | 27.86 | 22.44 | 55.60 | 20.67 | 48.90 | 36.54 |
| | | 27.16 | 0 | 53.27 | 49.00 | 24.74 | 22.01 | 56.42 | 19.26 | 55.20 | 39.25 |
| +UL | 256 | 27.87 | 0 | 51.46 | 56.00 | 27.27 | 21.75 | 56.20 | 20.77 | 34.70 | 37.00 |
| | | 28.32 | 0 | 50.82 | 56.00 | 28.03 | 22.18 | 56.63 | 20.77 | 34.10 | 37.10 |
| | | 27.93 | 0 | 50.74 | 54.00 | 27.94 | 21.24 | 55.05 | 20.16 | 33.90 | 36.37 |
| | | 27.98 | 0 | 50.82 | 54.00 | 28.03 | 21.58 | 55.87 | 20.80 | 33.80 | 36.37 |
| | | 28.03 | 0 | 51.14 | 54.00 | 27.06 | 22.44 | 55.60 | 20.56 | 38.70 | 37.19 |
| OPT$_{2.7B}$ | - | 45.84 | 63.57 | 61.01 | 77.00 | 60.77 | 26.88 | 73.83 | 23.85 | 60.80 | 53.74 |
| +DeMEM | - | 41.57 | 53.73 | 60.22 | 76.00 | 58.08 | 24.74 | 72.41 | 23.71 | 60.90 | 52.20 |
| +UL | 32 | 30.88 | 0.75 | 52.56 | 58.00 | 29.50 | 23.20 | 57.61 | 20.67 | 55.60 | 41.00 |
| | | 25.32 | 0 | 50.35 | 50.00 | 24.20 | 22.26 | 54.46 | 19.59 | 55.20 | 37.67 |
| | | 28.26 | 0 | 51.14 | 51.00 | 25.42 | 23.37 | 54.89 | 20.36 | 55.20 | 38.70 |
| | | 25.28 | 0 | 51.69 | 49.00 | 23.82 | 21.84 | 54.46 | 19.19 | 55.20 | 37.56 |
| | | 25.37 | 0 | 50.19 | 53.00 | 24.45 | 22.86 | 54.57 | 19.09 | 55.20 | 38.09 |
| +UL | 128 | 27.31 | 0 | 51.93 | 47.00 | 24.70 | 22.01 | 57.39 | 19.56 | 55.20 | 38.14 |
| | | 37.09 | 37.55 | 49.88 | 65.00 | 39.39 | 22.61 | 64.09 | 20.83 | 54.50 | 44.17 |
| | | 27.67 | 0 | 51.85 | 47.00 | 24.70 | 22.18 | 56.36 | 19.09 | 55.20 | 38.00 |
| | | 36.16 | 34.32 | 49.64 | 64.00 | 38.88 | 21.75 | 62.84 | 21.84 | 54.40 | 43.69 |
| | | 31.67 | 9.64 | 49.48 | 62.00 | 31.14 | 23.12 | 59.57 | 22.17 | 51.40 | 41.32 |
| +UL | 256 | 25.89 | 0 | 51.06 | 49.00 | 24.36 | 22.61 | 56.63 | 19.36 | 55.20 | 38.01 |
| | | 27.21 | 0 | 52.56 | 43.00 | 24.70 | 22.35 | 56.96 | 19.32 | 55.20 | 37.66 |
| | | 31.10 | 7.97 | 50.27 | 60.00 | 31.52 | 22.61 | 60.44 | 22.11 | 41.90 | 39.99 |
| | | 26.86 | 0 | 50.43 | 44.00 | 24.24 | 21.67 | 55.76 | 19.43 | 55.20 | 37.20 |
| | | 25.32 | 0 | 50.59 | 54.00 | 24.62 | 22.44 | 54.18 | 18.79 | 55.20 | 38.14 |

Table 5: Main Results: OPT averaged 5 random samples (s = 32, 128, and 256) for UL. UL = knowledge unlearning, DeMEM = DeMemorization. LM ACC = average accuracy of 8 classification datasets. Lambada Accuracy is excluded from the average due to anomalies

| Model | #Samples | Lamba. (PPL)↓ | Wikitext. (PPL)↓ |
|---|---|---|---|
| NEO$_{125M}$ | - | 30.26 | 32.28 |
| +DeMEM | - | 33.58 | 33.13 |
| +UL | 32 | 10919.67 | 357.79 |
| | | 1818857.11 | 3961.67 |
| | | 7405.89 | 335.80 |
| | | 3385138.77 | 6732.11 |
| | | 25647.89 | 144102.93 |
| +UL | 128 | 2655013035093.51 | 9621014 |
| | | 124900785 | 36560950 |
| | | 182274.05 | 1935.31 |
| | | 1395018915.85 | 163375.73 |
| | | 747238174142.82 | 2072110.35 |
| +UL | 256 | 128824105.70 | 40390.67 |
| | | 17736.35 | 41620.78 |
| | | 5446764.91 | 9477.52 |
| | | 47724404.48 | 22130.00 |
| | | 9320659.24 | 12115.23 |
| NEO$_{1.3B}$ | - | 7.49 | 16.16 |
| +DeMEM | - | 9.01 | 16.70 |
| +UL | 32 | 31.33 | 26.77 |
| | | 20.60 | 24.67 |
| | | 6.61 | 22.39 |
| | | 7.53 | 20.90 |
| | | 7.087 | 27.16 |
| +UL | 128 | 747.21 | 53.23 |
| | | 14.52 | 36.09 |
| | | 4920762.54 | 770.51 |
| | | 41.42 | 41.93 |
| | | 342.03 | 41.51 |
| +UL | 256 | 13.72 | 61.20 |
| | | 189789.40 | 227.06 |
| | | 189367.90 | 91.03 |
| | | 681965.60 | 171.54 |
| | | 705.34 | 42.59 |
| NEO$_{2.7B}$ | - | 5.62 | 13.93 |
| +DeMEM | - | 6.51 | 14.15 |
| +UL | 32 | 6.13 | 19.87 |
| | | 2992343.20 | 1531.31 |
| | | 10.44 | 28.07 |
| | | 10.28 | 17.41 |
| | | 6.23 | 17.11 |
| +UL | 128 | 17.83 | 61.20 |
| | | 41.04 | 63.78 |
| | | 8.91 | 16.15 |
| | | 10.58 | 33.69 |
| | | 53.39 | 116.89 |
| +UL | 256 | 25.89 | 0 |
| | | 27.21 | 0 |
| | | 31.10 | 7.97 |
| | | 26.86 | 0 |
| | | 25.32 | 0 |

Table 6: Perplexity Results On Lambada & Wikitext: NEO averaged 5 random samples (s = 32, 128, and 256) for UL. UL = knowledge unlearning, DeMEM = DeMemorization.

| Model | #Samples | Lamba. (PPL)↓ | Wikitext. (PPL)↓ |
|---|---|---|---|
| OPT$_{125M}$ | - | 26.02 | 31.94 |
| +DeMEM | - | 31.14 | 35.35 |
| OPT$_{1.3B}$ | - | 6.64 | 16.41 |
| +DeMEM | - | 7.61 | 17.39 |
| OPT$_{2.7B}$ | - | 5.11 | 14.31 |
| +DeMEM | - | 7.61 | 15.25 |

Table 7: Perplexity Results On Lambada & Wikitext: OPT For Original LM & DeMEM Since UL produced Infinity.

## F Baseline Method Hyperparameters

We selected the hyperparameters for UL based on (Jang et al., 2022) for NEO models, using the number of epochs required for unlearning until the target sequences meet the forgetting criteria. For OPT models, we used half the number of epochs compared to NEO models in specific sizes, as OPT models achieved the same loss as NEO models but in fewer epochs.

## G Memorization's Assumptions

As previously discussed, presenting assumptions to address the memorization problem often leads to incomplete solutions. This is evident in the case of differential privacy, which assumes whether the data is private or not. Similarly, UL assumes that the training and evaluation data are memorized, which is impractical in real-world applications considering that language models are trained on vast corpora with billions of tokens. Furthermore, fine-tuning an LM in an application involving potentially sensitive/private data poses challenges in splitting the data into sensitive/private and non-sensitive/private portions for the purpose of forgetting (Levine, 2021; Porcaro, 2022; Brown et al., 2022). On the other hand, DeMemorization does not rely on assumptions about the training data that need to be unlearned. Instead, we fine-tune the LM to learn a universal policy that reduces the relationship between the prefix and suffix. This policy achieves its objective by replacing the token with a similar entity or a context that is semantically correct but not directly linked to the same prefix, as illustrated in Figure 3. Another assumption is the limited number of samples to be unlearned at once, which we discussed before.

## H Hardware & Software Dependencies

In order to fine-tune GPT-Neo models of sizes 125M and 1.3B, we utilized a cluster of two V100 GPUs, each equipped with 32GB of VRAM. The 125M model required approximately 0.38 minutes per PPO epoch, resulting in a total computation time of 3.04 minutes for six epochs. The 1.3B model required a slightly longer computation time of 1.68 minutes per PPO epoch, for a total of 13.44 minutes over eight epochs. For the largest variant, GPT-Neo 2.7B, we utilized a cluster of four V100 GPUs, each with 32GB of VRAM, and employed a sharding strategy with zero 3 (Rasley et al., 2020). Each PPO epoch for this model required 5.125 minutes, resulting in a total computation time of approximately 20 minutes over four epochs. For finetuning those models, we employed the HuggingFace library (Wolf et al., 2019) for training and Pytorch (Paszke et al., 2017) for parallelizing the model. For RL fine-tuning, we employed TRL (Transformer Reinforcement Learning) library(von Werra et al., 2020).