# OpenReview forum: "Preserving Privacy Through Dememorization:  An Unlearning Technique For Mitigating Memorization Risks In Language Models"
_EMNLP/2023/Conference — EMNLP 2023 Main_

### Official Review · Reviewer_yMwJ · 2023-08-04

**Soundness:** 2

**Excitement:**

3: Ambivalent: It has merits (e.g., it reports state-of-the-art results, the idea is nice), but there are key weaknesses (e.g., it describes incremental work), and it can significantly benefit from another round of revision. However, I won't object to accepting it if my co-reviewers champion it.

**Paper Topic And Main Contributions:**

This paper focuses on the problem of memorization issue in LLM. In this paper, the author proposes a Dememorization framework for mitigating memorization by using a reinforcement learning feedback approach with just a few parameter updates. In detail, this paper uses a prefix as input for the language model, and computes the negative BERTScore to dissimilarity score between the true suffix and generated suffix, which is then used as a reward signal to maximize in training.

**Reasons To Accept:**

1. The paper is well-written for the most part and easy to follow.
2. The experiments are organized well in effectiveness of the Dememorization, the impact of increasing the context length and general language model capabilities of Dememorized model on downstream tasks .

**Reasons To Reject:**

1. The title mentioned the Unlearning Technique, but the method part did not give too many relevant descriptions, or highlight your own innovation points.  The methods section is too simple and doesn't focus on what the title is supposed to show.
2. The authors do not give a comprehensive study of the results and the comparison with other papers or baselines. There is some confusion in tables. For example, in the header of table 1, LM(PPL) goes down, but the actual value goes up. I'm not sure what the down arrow means, is it performance or number?
3. Should pay more attention to design the layout of tables.
4. No theoretical proof or guarantee to prove that the data is really be forgotten.

**Reproducibility:**

3: Could reproduce the results with some difficulty. The settings of parameters are underspecified or subjectively determined; the training/evaluation data are not widely available.

**Reviewer Confidence:**

2: Willing to defend my evaluation, but it is fairly likely that I missed some details, didn't understand some central points, or can't be sure about the novelty of the work.

---

> ### Author Rebuttal · Authors · 2023-08-28
>
> Thank you, reviewer yMwJ, for reviewing our paper. Please consider raising your score if we are able to address some of your concerns. Please find the responses for your feedback below:
>
> >The title mentioned the Unlearning Technique, but the method part did not give too many relevant descriptions or highlight your own innovation points.
>
> Starting from lines 137 to 143, we define the innovations and findings of this research paper:
> 1. Our method achieves comparable results to the baselines in the literature from a privacy wise while offering better accuracy in the performance of general capabilities of the language model.
>
> 2. The recent solutions try to replace the true suffix with any other sequence of tokens, even if it is not related to the context of the sentence, which affects the capabilities of the pre-trained model. However, in our method, we tried to tackle this problem using a paraphrasing approach to avoid memorization and simultaneously output a meaningful sequence related to the context of the sentence.
>
> 3. Our method is unaffected by changing the number of samples to be unlearned, as the previous method hurts the general LM capabilities when increasing the number of protected samples.
>
> 4. The proposed method can generalize to unseen samples, unlike the recent unlearning methods that are required to train and evaluate on the same samples. As a result, it requires training multiple times for new samples to be unlearned.
>
> > The methods section is too simple and doesn't focus on what the title is supposed to show.
>
> The methodology section starts by describing how we achieve forgetting or dememorization using the dissimilarity policy. In the first subsection, 4.1, we introduce the RL settings and basis for our approach, then demonstrate the chosen reward function to achieve our goal of unlearning. Finally, we present the hyperparameters and the convergence. In the next subsection, 4.2, we define our concept of memorization, which is the approximate memorization, and then what metric we choose to quantify it. Finally, we define the definition of forgetting from our perspective as minimizing the relationship between the prefix and suffix. Can you please point out the part that needs more clarification so we can include it in the revision?
>
> >The authors do not give a comprehensive study of the results and the comparison with other papers or baselines.
>
> We want to direct your attention to two sections of the paper: Related Work from 249 to 260 and Section 5.1.3 Baseline Methods from 433 to 450, where we compared our proposed method with two other baselines: the first relied on preprocessing the data using deduplication, and the second focused on knowledge unlearning (UL method), which is the only method present in the literature for addressing memorization using the unlearning approach. Additionally, we included a detailed comparison discussion of the proposed and deduplication methods' performance in section 5.2.2 of the paper.
>
> >There is some confusion in tables. For example, in the header of table 1, LM(PPL) goes down, but the actual value goes up. I'm not sure what the down arrow means, is it performance or number?
>
> We are sorry for any confusion caused by the clarity in the tables; the arrow goes down means lower values are better, while the arrow goes up means higher values are better. Thanks for the suggestion, We will add this clarification in our revision.
>
> >Should pay more attention to design the layout of tables.
>
> Thank you for your suggestion, but can you please point out the problems in the layout other than the ones that you mentioned before in order to edit them in the revised version?
>
> >No theoretical proof or guarantee to prove that the data is really be forgotten.
>
> The forgetting is measured by the similarity metric “SacreBLEU,” which we used based on empirical justification proposed by [1] to capture the approximate memorization. As shown in Tables 1 and 2, negative sacrebleu(the metric that measures forgetting) is lower in the pre-trained language model before applying the dememorization or UL approaches, indicating higher memorization and lower forgetting. While after employing the dememorization or UL, it shows higher negative sacrebleu, indicating lower memorization and higher forgetting.
>
> References:
>
> [1]. Ippolito, Daphne, et al. "Preventing verbatim memorization in language models gives a false sense of privacy." arXiv preprint arXiv:2210.17546 (2022).

---

### Official Review · Reviewer_N84C · 2023-08-07

**Soundness:** 2

**Excitement:**

2: Mediocre: This paper makes marginal contributions (vs non-contemporaneous work), so I would rather not see it in the conference.

**Paper Topic And Main Contributions:**

This paper proposes a method to mitigate privacy risks in language models. Given a pre-trained language model, a Reinforcement learning approach is proposed where the model is encouraged to produce continuations dissimilar to ground-truth continuations. The proposed method is shown to be more effective at addressing privacy risks while minimally affecting the capabilities of the pre-trained model.


**Questions For The Authors:**

* What data is used for the proposed dememorization approach? This was not made clear.
* Eq 4: BLEU or SacreBLEU?


**Reasons To Accept:**

* Improving privacy properties of language models is an important problem
* Method works better than the baseline UL method on some metrics
* Method is more robust to number of samples to be unlearned and increase in prompting context length


**Reasons To Reject:**

* Poor clarity (see further comments in Presentation suggestions)
* Scope of the work seems limited. Most of the experiments offer comparisons against only a single baseline approach (UL).
* The intuition and motivations for the proposed approach are not clearly presented. The ‘12 red street’ example that repeatedly appears in the paper is not clear/convincing.
* Description of the proposed approach (sec 4.1) is superficial and missing technical details. I could only get a surface level understanding of the method.
  - paragraph in line 267 talks about an ‘environment’ which is vague and unclear.
  - Mathematical descriptions of the method are missing (e.g., KL penalty, value network)
  - What is the loss function?
* Missing Ablation: There was no empirical justification for using the SacreBLEU metric for computing dissimilarity.
* While the proposed method harms the pre-trained model less, it is also worse in terms of the privacy metric (SacreBLEU). As a result, it is not clear to me if the proposed method is more favorable. Table 1 is also hard to interpret. There is no highlighting of best numbers for each metric.
* Given that most comparisons are focused on UL, I would have expected to see a proper technical description of UL as a preliminary. Further, UL is the only baseline considered and there are no comparisons against other baselines.


**Reproducibility:**

1: Could not reproduce the results here no matter how hard they tried.

**Reviewer Confidence:**

3: Pretty sure, but there's a chance I missed something. Although I have a good feel for this area in general, I did not carefully check the paper's details, e.g., the math, experimental design, or novelty.

**Typos Grammar Style And Presentation Improvements:**

* Writing/clarity needs significant improvement.
* Line 125: “explicit, implicit assumptions’ -> why not just say ‘assumptions’?
* Sentence in line 171 is unclear
* The paragraph that starts at line 249 seems out of place. The paper is plagued by such structure issues.
* SacreBLEU is claimed as a widely used metric but was not cited
* line 378: ‘designed to be easy to extract to assess’ - what does this mean?
* line 390: ‘100 prefix tokens, 50 prefix tokens’ -> unclear
* line 399: ‘pre-prefix’ -> what is this
* All the tables are very small and hardly readable
* 5.1.3: Make it clear that OPT was trained on deduplicated data
* line 456 has a long sentence which is hard to comprehend
* 5.1.5: MA was never defined
* Table 1 is referenced in page 7 but the table appears in page 5

---

> ### Author Rebuttal · Authors · 2023-08-28
>
> Thank you, reviewer N84C, for reviewing our paper. Please consider raising your score if we are able to address some of your concerns. Please find the responses for your feedback below:
>
> >Scope of the work seems limited. Most of the experiments offer comparisons against only a single baseline approach (UL).
>
> We want to direct your attention to two sections of the paper: Related Work from 249 to 260 and Section 5.1.3 Baseline Methods from 433 to 450, where we compared our proposed method with two other baselines: the first relied on preprocessing the data using deduplication, and the second focused on knowledge unlearning (UL method), which is the only method present in the literature for addressing memorization using the unlearning approach. Additionally, we included a detailed comparison discussion of the proposed and deduplication methods' performance in section 5.2.2 of the paper.
>
> >The intuition and motivations for the proposed approach are not clearly presented. The ‘12 red street’ example that repeatedly appears in the paper is not clear/convincing.
>
> We aim to address the memorization risk of language models. The recently used approach relies on removing the data points entirely from the model’s parameters. For example, if we have a sentence in the training data, “John lives in Canada.” the suffix “John lives” and the prefix “in Canada.” The memorization problem arises when we give the model the prefix “John lives” and the model outputs “in Canada.” The recent solution tries to replace the true suffix with any other sequence of tokens, even if it is not related to the context of the sentence, which affects the capabilities of the pre-trained model. However, in our method, we tried to tackle this problem using a paraphrasing approach to avoid memorization and, simultaneously output a meaningful sequence related to the context of the sentence. To further illustrate, if we give the input prefix “John lives,” the dememorized model should output “in USA.” or “in Australia.” or any other suffix that is related to the input prefix. Therefore, we define memorization as a relationship between the prefix and the suffix and try to minimize the relationship between them. Consequently, we quantify this relationship using the sacre-bleu metric. The previous statements are mentioned in lines 105–112 and 337–360 in the paper.
>
> >paragraph in line 267 talks about an ‘environment’ which is vague and unclear.
>
> The core component of our approach is using reinforcement learning to fine-tune the language model to a policy that does not memorize sentences. In the context of reinforcement learning (RL), the "environment" is essentially the place or context where an RL agent operates and learns. Think of it as a virtual or simulated world where the agent interacts to learn how to perform tasks. This environment contains all the elements and rules that the agent needs to understand in order to achieve its goals. In our case, the agent is the generative model that takes some actions that can affect the environment; therefore, the learned policy. For the model to converge to the desired optimized policy that meets our goals and objectives, we use the reward function (sacrebleu), which serves as a way to measure the quality of the agent's actions and decisions. It quantifies the desirability of the agent's behavior. The agent's goal is to find a policy (a strategy for selecting actions in different states) that maximizes the expected cumulative reward over time. The preceding statements are mentioned in lines 268-288 of the paper.
>
> >Mathematical descriptions of the method are missing (e.g., KL penalty, value network)
>
> KL penalty, or value network, are the basis of reinforcement learning (RL) training setups. Therefore, we did not feel the need to include the mathematical equations because they are commonly used in every RL task. However, That is a good suggestion. We will say more about this in our revision.
>
> >What is the loss function?
>
> In terms of reinforcement learning (RL), there is no loss function per se. However, it does have a concept similar to a loss function, although it is typically referred to as the "reward function." In reinforcement learning, the goal is to teach an agent to make sequential decisions in an environment to maximize some notion of cumulative reward. The incorporated reward function in our method is defined in lines 281-288 and equation 3 of the paper, Which is the negative BERTScore combined with the KL penalty.
>
> >Missing Ablation: There was no empirical justification for using the SacreBLEU metric for computing dissimilarity.
>
> The justification for employing the SacreBLEU metric is based on following the work proposed by [1], which used SacreBLEU to measure the approximate memorization and proved its efficiency. Therefore, we employed it as we adopted the concept of approximate memorization. Thus, we used the negative value of the same metric to measure the dissimilarity because it initially measures the similarity. The previous statements are present in section 2.1, lines 175-182, and section 4.2, lines 319-335.
>
> >While the proposed method harms the pre-trained model less, it is also worse in terms of the privacy metric (SacreBLEU). As a result, it is not clear to me if the proposed method is more favorable.
>
> Providing stronger privacy protections for LMs may become meaningless if it requires sacrificing their original capabilities. Thus, it’s a tradeoff. We cannot only rely on achieving high performance in terms of privacy metrics and neglect the quality of the generated suffix, the performance of the LM, or how related it is to the original prefix. However, our method combined with the deduplication method achieves comparable results to the UL method with slight differences while offering better accuracy.
> Since data deduplication is applied in most of the recent & large language models [2,3,4,5,6,7], we believe our approach combined with deduplication will effectively mitigate memorization while achieving insignificant degradation(∼0.5%) in the Language model performance. So, we ensure that the generated suffixes are related to the context of the prefix and, at the same time, achieve high forgetting rates. The previous statements and results can be found in Section 5.2.2 and Table 2 of the paper.
>
> >Given that most comparisons are focused on UL, I would have expected to see a proper technical description of UL as a preliminary. Further, UL is the only baseline considered and there are no comparisons against other baselines.
>
> Recalling our previous responses, we compare our method with two baselines, and UL is one of them. Additionally, we included a brief description of the approach, its main components, advantages, and limitations. However, we will try to incorporate further information about its details in the revision.
>
> >What data is used for the proposed dememorization approach? This was not made clear.
>
> We employed a subset of the pile dataset for training. Then, we evaluated and tested the model on other subsets of the pile dataset. So, the training and testing are done on completely different sets of data. Each sample consists of a 50-token prefix and a 50-token suffix; the model takes the prefix as input and then generates a suffix, which is compared to the original suffix using BERTScore. The training set is 13,500 samples, while the test set is 1,500 samples. All the details regarding the data employed in our approach are presented in Section 5.1.1. Can you please further clarify which part of the dataset was not clear in order to answer your question?
>
> >Eq 4: BLEU or SacreBLEU?
>
> Mathematically, both BLEU and SacreBLEU are the same. The only difference is from a technical perspective, as SacreBLEU provides a more robust implementation, offering better handling of tokenization, corpus-level processing, and support for multiple languages.
>
> >Writing/clarity needs significant improvement.
>
> Thank you for your suggestion, but can you please point out the sections other than the ones that you mentioned before in order to revise them in the revised version?
>
> >Line 125: “explicit, implicit assumptions’ -> why not just say ‘assumptions’?
>
> To precisely address what assumptions we mean, we use “explicit” or “implicit” assumptions since they are different.
>
> >The paragraph that starts at line 249 seems out of place. The paper is plagued by such structure issues.
>
> This paragraph summarizes the limitations we found in the literature and how our proposed method addresses them. That’s why it’s included at the end of the related work section.
>
> >line 378: ‘designed to be easy to extract to assess’ - what does this mean?
>
> This means that the designed dataset is constructed in such a way that the Langauge model can memorize it, so we can assess how the unlearning techniques perform. Since if we used samples that LM does not memorize, we wouldn’t be able to assess the efficiency of the unlearning approaches.
>
> >line 390: ‘100 prefix tokens, 50 prefix tokens’ -> unclear
>
> In the paper, this line is written as follows:
> “Each sample consists of a 200-token sequence divided into 100 pre-prefix tokens, 50 prefix tokens, and 50 suffix tokens”. So, it’s pre-prefix, not prefix only.
>
> >line 399: ‘pre-prefix’ -> what is this
>
> The base components of the sample in the data are “prefix” and “suffix.” We further investigated using a longer context to evaluate how the model would perform. Therefore, we added another component to the sample, which is “pre-prefix,” which denotes the 100 tokens preceding the original prefix. Thus, the components of each training sample in the longer context setups are (pre-prefix, prefix, and suffix).
>
> >"Typos Grammar Style And Presentation Improvements"
>
> Thank you for the detailed suggestions. We will incorporate them in our revision.
>
> References:
>
> [1]. Ippolito, Daphne, et al. "Preventing verbatim memorization in language models gives a false sense of privacy." arXiv preprint arXiv:2210.17546 (2022).
>
> [2]. Penedo, Guilherme, et al. "The RefinedWeb dataset for Falcon LLM: outperforming curated corpora with web data, and web data only." arXiv preprint arXiv:2306.01116 (2023).
>
> [3]. Touvron, Hugo, et al. "Llama: Open and efficient foundation language models." arXiv preprint arXiv:2302.13971 (2023).
>
> [4]. Biderman, Stella, et al. "Pythia: A suite for analyzing large language models across training and scaling." International Conference on Machine Learning. PMLR, 2023.
>
> [5]. Taylor, Ross, et al. "Galactica: A large language model for science." arXiv preprint arXiv:2211.09085 (2022).
>
> [6]. Scao, Teven Le, et al. "Bloom: A 176b-parameter open-access multilingual language model." arXiv preprint arXiv:2211.05100 (2022).
>
> [7]. Black, Sid, et al. "Gpt-neox-20b: An open-source autoregressive language model." arXiv preprint arXiv:2204.06745 (2022).

---

### Official Review · Reviewer_mR2L · 2023-08-07

**Soundness:** 3

**Excitement:**

3: Ambivalent: It has merits (e.g., it reports state-of-the-art results, the idea is nice), but there are key weaknesses (e.g., it describes incremental work), and it can significantly benefit from another round of revision. However, I won't object to accepting it if my co-reviewers champion it.

**Paper Topic And Main Contributions:**

The paper describes a finetuning method based on reinforcement learning in order to de-memorize
potentially privacy-sensitive information; that is, prevent a generating model from completing text
like "most most cherished secret is " with the actual completion of such an instance that was used
in training. At the core of the technique to use a reward for completing text sequences that
*differ* from the actual training text instances as measured by a (di)similarity score (in this
case as determined by BertSCORE, a similarity metric based on token embeddings in a BERT model).
The method also includes a penalty (details not certain) to prevent the fine-tuning from deviating
too much from the original. Together these goals are nicely summarized as training a model not to
generate but to "paraphrase".

The paper experimentally compares itself against one "knowledge unlearning" (UL) baseline that
targets specific sets of tokens to discount in finetuning. Experiments show that the proposed
method offers a better accuracy vs. privacy tradeoff in that while it may not be as effective at
unlearning, it significantly better preserves the performance of the model. The UL technique is
shown to breakdown for sufficiently large sets of samples.

**Questions For The Authors:**

- Results for performance under DeMEM are reported as one number across all values for #Samples but
  different values for memorization score. This is a bit confusing as it is implying there is only
  one fine-tuning process independent of the number of samples. Was DeMEM not fine-tuned with the
  same samples as UL?

- My list of reasons to reject includes some important questions whose answer will have a big
  impact on my evaluation. Please answer those.

- What is "the KL penalty" ?

- Line 427 discusses not reporting anomalous results but the reason of why the should be considered
  anomalous is not provided. Please explain.

- Infinite results suggest numerical instability. One place I could see this happening is if the
  usual small epsilon was not correctly employed to control numerical singularities. Did the
  evaluation not include such techniques? Related, at what level of perplexity does the resulting
  model produce unrecognizable gibberish? I presume it is at a point far below the levels being
  reported, even the non-infinite ones.

**Reasons To Accept:**

+ "Paraphrasing" as a goal is an intuitive goal that avoids memorization and perhaps even other
  undesired outcomes in NLP ML.

+ Reasonable results on multiple LLMs and unlike UL, offering usable models while performing some
  amount of unlearning.

**Reasons To Reject:**

- It seems that the methodology aims at unlearning particular instances and evaluates that
  unlearning on those instances that were unlearned. That is, no sort of generalization of the idea
  of "paraphrase" is tested. The setting is thus: I have a pre-trained model and I know it
  generates a set of instances S based on their prefix. I don't want it to generate those exact
  instances but something similar. In this case there may be simpler things to do than fine-tune.
  For example: detect instances of S and output completions generated by a different model (or just
  refuse to complete). Alternatively, if S are known ahead of training, remove them from the
  training data.

  Suggestion: First: is my reading of the evaluation and goals accurate? If so, include some
  discussion as to why fine-tuning should be used to address specific instance as opposed
  instance-specific handling (example above). If my reading is not correct, please describe and
  evaluate (if not already) in the work how the unlearning process generalizes to non unlearned
  instances.

UPDATE: See also comment added post-rebuttal.

- The comparison to UL may be unfair.

  Firstly, the presented method includes:

      "KL penalty is applied per token using a reference model (the pre-trained model before
       finetuning.) This prevents the fine-tuned model from generating a suffix that deviates too
       much from the reference language model"

  This likely prevents the model degradation that UL exhibits. Could the same be used in UL?
  Related, I feel like the use of this "KL penalty" (at least based on its stated goals) is
  significant enough that it should be discussed in more detail.

  Suggestion: Discuss whether similar approach to KL penalty can be enabled for UL and if so, what
  would the experimental results look like?

UPDATE: See also comment added post-rebuttal.

  Second, there is sufficient room between in memorization results for UL as compared to the
  proposed method to tune parameters for alternate tradeoffs. That is, tune UL to achieve the same
  memorization score and compare the resulting performance score. I suspect the number of samples
  conceptually cannot be tuned as they are also the ones being tested but this still leaves the
  ability to only unlearn a smaller than the number of samples instances and take the resulting
  penalty in the memorization score for the others.

  Suggestion: Attempt to fine-tune UL for a comparison of one metric on the same level of the
  other. Alternatively discuss why this is not possible.

Smaller points:

- Intro includes the term "PPO" but it is not explained until later in the paper. Please include a
  brief definition at first use.

- Some of the qualitative results in Figure 6 would be nice to include in the main paper.

**Reproducibility:**

3: Could reproduce the results with some difficulty. The settings of parameters are underspecified or subjectively determined; the training/evaluation data are not widely available.

**Reviewer Confidence:**

2: Willing to defend my evaluation, but it is fairly likely that I missed some details, didn't understand some central points, or can't be sure about the novelty of the work.

**Typos Grammar Style And Presentation Improvements:**

- Equation 4 has S_G twice. Related, the definition is a bit vague around the "precision" part. If
  the definition is important, include more details. If not important, consider removing.

- Grammar issue near "to not hurts".

- Grammar issue near "OPT LMs fine-tuned them".

- Grammar issue near "became more robust privacy LMs".

- This sentence:

    "We employed a subset of the Pile dataset, which was released as a benchmark for training data
     extraction attacks on large Language Models."

  Make it sound like Pile was the benchmark for data extraction attacks. Please rephrase.

- Figure 2 has text which is far too small to be readable. Figure 3 is better but still too small
  to read.

- Why is "GENERATION AS A TOKEN-LEVEL MDP" capitalized?

- Syntax/grammar issue near "knowledge. as a". Also, the statement made there is not clear:

    "a longer context in a language model can be considered a form of attack"

  You mean a longer context can be used to carry out a more effective attack?

---

> ### Author Rebuttal · Authors · 2023-08-28
>
> Thank you, reviewer mR2L, for reviewing our paper. Please consider raising your score if we are able to address some of your concerns.
> Please find the responses for your feedback below:
>
> > The paper experimentally compares itself against one "knowledge unlearning" (UL) baseline that targets specific sets of tokens to discount in fine-tuning.
>
> We would like to direct your attention to two sections of the paper: Related work from 249 to 260 and Section 5.1.3 Baseline Methods from 433 to 450, where we compared our proposed method with two other baselines: the first relied on preprocessing the data using deduplication, and the second focused on knowledge unlearning (UL method), which is the only one present in the literature. Additionally, we included a detailed comparison discussion of the proposed and deduplication methods' performance in section 5.2.2 of the paper.
>
> > Experiments show that the proposed method offers a better accuracy vs. privacy tradeoff in that while it may not be as effective at unlearning, it significantly better preserves the performance of the model.
>
> That is correct; however, providing stronger privacy protections for LMs may become meaningless if it requires sacrificing their original capabilities. Thus, it’s a tradeoff. We cannot only rely on achieving high performance in terms of privacy metrics and neglect the quality of the generated suffix, the performance of the LM, or how related it is to the original prefix. However, our method combined with the deduplication method achieves comparable results to the UL method with slight differences while offering better accuracy. Since data deduplication is applied in most of the recent & large language models (Penedo et al., 2023; Touvron et al., 2023; Biderman et al., 2023; Taylor et al., 2022; Scaoet al., 2022; Black et al., 2022), we believe our approach combined with deduplication will effectively mitigate memorization while achieving insignificant degradation(∼0.5%) in the Language model performance. So, we ensure that the generated suffixes are related to the context of the prefix and, at the same time, achieve high forgetting rates. The previous statements and results can be found in Section 5.2.2 and Table 2 of the paper.
>
> >It seems that the methodology aims at unlearning particular instances and evaluates that unlearning on those instances that were unlearned. That is, no sort of generalization of the idea of "paraphrase" is tested. The setting is thus: I have a pre-trained model and I know it generates a set of instances S based on their prefix. I don't want it to generate those exact instances but something similar. In this case there may be simpler things to do than fine-tune. For example: detect instances of S and output completions generated by a different model (or just refuse to complete). Alternatively, if S are known ahead of training, remove them from the training data.
> Suggestion: First: is my reading of the evaluation and goals accurate? If so, include some discussion as to why fine-tuning should be used to address specific instance as opposed instance-specific handling (example above). If my reading is not correct, please describe and evaluate (if not already) in the work how the unlearning process generalizes to non unlearned instances.
>
> The methodology and evaluation setup that you mentioned are correct descriptions, but for the UL method, because they attempt to forget a specific number of samples (S) by training and then evaluating the model's unlearning on the exact samples (S). However, in our approach, we train the model on separate training data (mentioned in Section 5.1.1 from 389 to 404) and then evaluate the unlearning on the samples (S) used in the UL method to compare our results with theirs. Therefore, the test set is fixed in both methods, but the training data in our method is entirely different. Thus, our model generalizes because we train and test it on different sets of data. As a result, this is a strength point that proves the effectiveness of our method's generalizability against UL, not a weakness point. The previously mentioned details of the training/evaluation setup of the proposed method are described in sections 5.2.3 (573 to 585) and 5.1.4.
>
> >What is "the KL penalty" ?
>
> First, let us define the function of the KL penalty; it computes the KL divergence between the probability distributions of the generations of initialized LM and the finetuned LM. An important point to note is that the KL penalty is not a separate component from the core design of our reward function, as we designed the reward function to include two components: the dissimilarity score and the KL penalty. Each of the reward components has an important role; the dissimilarity score encourages the model to learn a policy that generates different samples from the original prefix, but we have to make sure that the policy does not significantly deviate the generations from the original; to achieve this, we included the KL penalty in the reward; thus, the generated and the original samples will have the same meaning. Employing the KL penalty as a component of the reward function is commonly used in most Reinforcement Learning tasks [1, 2, 3].
>
> >This likely prevents the model degradation that UL exhibits. Could the same be used in UL? Related, I feel like the use of this "KL penalty" (at least based on its stated goals) is significant enough that it should be discussed in more detail.
> Suggestion: Discuss whether similar approach to KL penalty can be enabled for UL and if so, what would the experimental results look like?
>
> If we recall the core detail of the UL approach, it is a method that relies on supervised learning, which means that the training of the model is done in a supervised manner as they maximize the loss of the language model, adding the KL penalty to their approach would completely change their method setup or objective and might impossible to compose it since we propose an RL method that uses a concept of reward function, not loss function. Their methods suffer from the degradation problem, so one of the solutions in our approach was to use the KL penalty along with the BERTScore in the reward In order to overcome it. Proposing the use of the KL penalty in their approach should be a further investigation for the paper's authors; our objective was to compare our results with theirs and overcome the limitations of their work. Again, this should be regarded as a strength point for our proposed method as opposed to the UL method. The details of using the KL penalty are mentioned in lines 299 to 304; however, thank you for your suggestion. We will include more details about it in our revision since it overcomes a considerable limitation of the previous works in the literature.
>
> >Second, there is sufficient room between in memorization results for UL as compared to the proposed method to tune parameters for alternate tradeoffs. That is, tune UL to achieve the same memorization score and compare the resulting performance score. I suspect the number of samples conceptually cannot be tuned as they are also the ones being tested but this still leaves the ability to only unlearn a smaller than the number of samples instances and take the resulting penalty in the memorization score for the others.
> Suggestion: Attempt to fine-tune UL for a comparison of one metric on the same level of the other. Alternatively discuss why this is not possible.
>
> If we correctly understand your question, we used the official hyperparameters proposed by the authors and are already tuned by them to achieve their objective. Since changing the values of the official hyperparameters will change the policy that their models learned. However, we tried to tune some of these hyperparameters; for example, reducing the number of epochs drastically decreased the dissimilarity score and increased the performance. Additionally, a core component of their method is called "forgetting threshold," which is their empirical definition of forgetting as they consider a token sequence to be forgotten and unsusceptible from extraction attacks when it meets their suggested threshold. Thus, we must keep these tuned hyperparameters, or we will oversight their whole approach. Further details of their approach can be found at [4].
>
> >Results for performance under DeMEM are reported as one number across all values for #Samples but different values for memorization score. This is a bit confusing as it is implying there is only one fine-tuning process independent of the number of samples. Was DeMEM not fine-tuned with the same samples as UL?
>
> As mentioned in the preceding responses, we finetune our model with separate training data and then evaluate it on various test sets. So, we only finetune the model once and then use it for evaluation on each test set. This proves the generalization of the learned policy, as it is independent of the number of protected samples. The details of this part can be found in lines 140 to 141 and 573 to 581.
>
> >Line 427 discusses not reporting anomalous results but the reason of why the should be considered anomalous is not provided. Please explain.
>
> The reason why they should be considered anomalous is provided in line 428: “as it shows such high values for perplexity & low value for accuracy for UL baseline.”  Including these values in the average of accuracy & perplexity will make the average score so low for accuracy & high for perplexity of the UL baseline but not for our approach, as it achieves good performance on this dataset, as reported in Appendix E. So for fair comparison we did not add them into the average performance of UL. However, all of the results are included in Appendix E.
>
> >Infinite results suggest numerical instability. One place I could see this happening is if the usual small epsilon was not correctly employed to control numerical singularities. Did the evaluation not include such techniques? Related, at what level of perplexity does the resulting model produce unrecognizable gibberish? I presume it is at a point far below the levels being reported, even the non-infinite ones.
>
> Can you please make the question more clear? Since I did not get what you mean by epsilon. As far as I know, there is no epsilon in equations in our approach and the same for UL. Also, there is no epsilon in the Perplexity equation or accuracy. Infinite and large values for perplexity happen in the UL baseline due to softens the probability distribution and leads to a more uniform distribution. The previous clarification is in lines 601 to 607. Additionally, it is provided by the authors of the paper.
>
> >Why is "GENERATION AS A TOKEN-LEVEL MDP" capitalized?
>
> To highlight the employed environment.
>
> >"a longer context in a language model can be considered a form of attack"
> You mean a longer context can be used to carry out a more effective attack?
>
> Yes, by this statement, we mean that a longer context can be used to carry out a more effective attack, as observed by [5] that employing a longer context increases the memorization percentage. We show the results of increasing the context length and how it can affect memorization in section 5.2.5.
>
> >"Typos Grammar Style And Presentation Improvements"
>
> Thank you for the detailed suggestions. We will incorporate them in our revision.
>
> References:
>
> [1]. Ramamurthy, Rajkumar, et al. "Is reinforcement learning (not) for natural language processing?: Benchmarks, baselines, and building blocks for natural language policy optimization." arXiv preprint arXiv:2210.01241 (2022).
>
> [2]. Ziegler, Daniel M., et al. "Fine-tuning language models from human preferences." arXiv preprint arXiv:1909.08593 (2019).
>
> [3]. Ouyang, Long, et al. "Training language models to follow instructions with human feedback." Advances in Neural Information Processing Systems 35 (2022): 27730-27744.
>
> [4]. Jang, Joel, et al. "Knowledge unlearning for mitigating privacy risks in language models." arXiv preprint arXiv:2210.01504 (2022).
>
> [5]. Carlini, Nicholas, et al. "Quantifying memorization across neural language models." arXiv preprint arXiv:2202.07646 (2022).

---

### Meta-Review · Area_Chair_zHSq · 2023-09-16

**Recommendation:** 3

**Metareview:**

Most reviewers agree that the current manuscript proposes a RL-based approach that leverages dissimilarity score between the authentic and generated completions as a reward signal. To enhance the paper attractive, the authors are encouraged to address several points: a) clearly articulate that the primary objective is to reduce memorization during pre-training phase in fine-tuning phase. 2) add more quantitative details in KL-divergence and UnLearning methods in a self-contained manner; 3) incorporate additional unlearning baselines if feasible. Otherwise, justifying why differential privacy methods cannot be applicable to pre-training of LLMs would provide more clarity.

---

### Decision · Program_Chairs · 2023-10-07

**Decision:**

Accept-Main

**Comment:**

Most reviewers agree that the current manuscript proposes a RL-based approach that leverages dissimilarity score between the authentic and generated completions as a reward signal. To enhance the paper attractive, the authors are encouraged to address several points: a) clearly articulate that the primary objective is to reduce memorization during pre-training phase in fine-tuning phase. 2) add more quantitative details in KL-divergence and UnLearning methods in a self-contained manner; 3) incorporate additional unlearning baselines if feasible. Otherwise, justifying why differential privacy methods cannot be applicable to pre-training of LLMs would provide more clarity.